# A comparison of straight-ray and curved-ray surface wave tomography approaches at near-surface studies

Mohammadkarim Karimpour[1], Evert Slob[2], Laura Valentina Socco[1]

[1]Politecnico di Torino, Turin, 10129, Italy
[2]Delft University of Technology, Delft, 2628 CN, Netherlands

*Correspondence to*: Mohammadkarim Karimpour (mohammadkarim.karimpour@polito.it)

**Abstract.** Surface waves are widely used to model shear-wave velocity of the subsurface. Surface wave tomography (SWT) has recently gained popularity for near-surface studies. Some researchers have used straight-ray SWT in which it is assumed that surface waves propagate along the straight line between receiver pairs. Alternatively, curved-ray SWT can be employed

by computing the paths between the receiver pairs using a ray-tracing algorithm. SWT is a well-established method in seismology and has been employed in numerous seismological studies. However, it is important to make a comparison between these two SWT approaches for near-surface applications since the amount of information and the level of complexity in near-surface are different from seismological studies. We apply straight-ray and curved-ray SWT to four near-surface examples and compare the results in terms of the quality of the final model and the computational cost. In three

examples we optimise the shot positions to obtain acquisition layout which can produce high coverage of dispersion curves. In the other example, the data have been acquired using a typical seismic exploration 3D acquisition scheme. We show that if the source positions are optimised, the straight-ray can produce S-wave velocity models similar to the curved-ray SWT but with lower computational cost than the curved-ray approach. Otherwise, the improvement of inversion results from curved-ray SWT can be significant.

**1 Introduction**

Surface waves are commonly analysed to build shear-wave velocity (VS) models. Surface wave tomography (SWT) is a well-established method in seismological studies, where numerous researchers have used it to construct subsurface velocity models at global and regional scale by inverting earthquake signals (Woodhouse and Dziewonski, 1984; Ekstrom et al., 1997; Ritzwoller and Levshin, 1998; Boschi and Dziewonski, 1999; Simons et al., 1999, Boschi and Ekstrom, 2002; Yao et

al., 2010). Some authors have applied SWT using ambient noise cross-correlation to retrieve regional crustal structures (Shapiro et al., 2004; Lin et al., 2008, Kästle et al., 2018).

SWT usually consists of three steps (Yoshizawa and Kennett, 2004; Yao et al., 2008). First, different path-averaged dispersion curves (DCs) are computed for different receiver pairs aligned with a source. Then, the DCs are inverted to build phase velocity maps at different period (frequency). Finally, the obtained phase velocity maps are inverted to produce 1D VS

models at different locations. However, the efficiency of SWT can be increased by the direct inversion of the path-averaged DCs, i.e., skipping the intermediate step of building phase velocity maps (Boschi and Ekstrom, 2002; Boiero, 2009; Fang et al., 2015).

Traditionally in seismology, SWT has been employed assuming great-circle propagation of surface waves (Trampert and Woodhouse, 1995; Ekstrom et al., 1997; Passier et al., 1997; Ritzwoller and Levshin, 1998; Boschi and Dziewonski, 1999;
van Heijst and Woodhouse, 1999; Simons et al., 1999; Boschi and Ekstrom, 2002; Lin et al., 2008; Yao et al., 2010; Bussat and Kugler, 2011; Kästle et al., 2018). However, some researchers have employed SWT not assuming the great-circle propagation of surface waves (Spetzler et al., 2002; Yoshizawa and Kennet, 2004; Lin et al., 2009). SWT has been used in seismological studies for decades and different SWT approaches have been compared by seismologists. For instance, Laske (1995) studied deviations from straight line in the propagation of long-period surface waves and concluded that they usually
have small effects on the propagation phase. Spetzler et al. (2001) applied both straight-ray and curved-ray SWT methods. They computed the maximum deviations of ray paths from straight lines and pointed out that this maximum is typically below the estimated resolution, except for long paths at short periods. Some studies showed that a more complex forward modelling in SWT did not improve the results (Sieminski et al., 2004; Levshin et al., 2005) while other studies reported obtaining better results (Ritzwoller et al., 2002; Yoshizawa and Kennett, 2004; Zhou et al., 2005). Trampert and Spetzler
(2006) pointed out that the choice of regularization has a major impact on SWT results. They studied SWT methods based on ray theory (straight-ray and curved-ray) and scattering theory in which the integral along the ray path is replaced by the integral over an influence zone. They showed that these methods are statistically alike and any model from one method can be obtained by the other one by changing the value of the regularization. They concluded that the only option to increase the resolution of the model is to increase and homogenize the data coverage. Bozdag and Trampert (2008) compared straight-ray
and curved-ray SWT methods in their study and mentioned that performing ray tracing could be so time-consuming that the potential gain in crustal corrections on a global scale might not be worth the additional computational effort imposed by ray tracing. Despite seismological studies, a comparison between the performance of straight-ray and curved-ray SWT at the near-surface scale is missing.

In near-surface studies, the shot locations can be optimised to ensure that a high coverage of DCs can be achieved. This abundance of information facilitates shallow 2D or 3D characterization with great details. Due to its ability to provide high lateral resolution, SWT has recently attracted the attention of researchers for near-surface studies, where high lateral heterogeneity is expected. Few researchers have used SWT for near-surface characterization assuming straight ray propagation of surface waves: Kugler et al. (2007) characterized shallow-water marine sediments using Scholte waves
dispersion data, Picozzi et al. (2009) applied SWT on high-frequency seismic noise data to construct a VS model up to 25 m depth, Rector et al. (2015) employed SWT to obtain a VS model in a mining site, Ikeda and Tsuji (2020) successfully applied SWT in  laterally heterogeneous media, Papadopoulou (2021) showed the applicability of SWT in near-surface

characterization in a mining site consisting of hard rocks, Khosro Anjom (2021) constructed a 3D VS model applying SWT on a large 3D dataset acquired for testing purposes in a mining area.

Since the level of complexity and lateral heterogeneity in the near-surface is expected to be higher than in most seismological studies, the straight ray approximation of surface waves may not be valid and curved-ray tomography should be used by means of ray tracing at each frequency. Fang et al. (2015) applied SWT on a shallow crustal study considering the effect of heterogeneity on wave propagation. They performed surface wave ray tracing at each frequency using a fast-marching method (Rawlinson and Sambridge, 2004). Wu et al. (2018) applied curved-ray SWT to obtain a shallow VS

model at a mining site. Barone et al. (2021) applied different tomography methods, including eikonal tomography, to 3D active seismic data.

In curved-ray SWT, at each iteration a ray tracing method is applied at each frequency component to compute the ray path between the source and receivers. Even though this can increase the accuracy of the final model, it will lead to higher computational cost compared with straight-ray SWT. The computational efficiency is of great importance in seismic near-

surface since, compared to seismological studies, the abundance of data at active seismic near-surface projects can increase the computational cost significantly. Therefore, it is necessary investigate the gained improvement together with the associated additional computational cost from the curved-ray SWT over the straight-ray approach.

We apply straight-ray and curved-ray SWT on four datasets. Two examples include 3D synthetic models containing lateral velocity heterogeneity. We then apply SWT on two field datasets to evaluate the method on real data. For each dataset, 3D

VS models from straight- and curved-ray SWT are obtained by direct inversion of DCs, and the accuracy and computational efficiency of the two approaches are compared.

## 2 Method

In this section, the applied methodology is described. Besides explaining the straight-ray and curved-ray SWT approaches and the differences between them, we also describe the employed procedure to optimise source positions and the process to

estimate the DCs from the raw data.

### 2.1 Optimisation of source layout

For a given (random or regular) array configuration, we can optimise the locations of shots to ensure having high coverage DCs with minimum number of shots based on the guidelines by Da Col et al. (2020). In this approach, many shot positions are defined as the potential shot candidates. For each shot, we find all receiver pairs aligned with the shot. After computing

all the possible receiver pairs for all the defined shot positions, the shots are sorted based on the number of in-line receiver pairs. Then we pick the shots which could provide the greatest number of unique pairs (i.e., potential DCs). If the data coverage is satisfactory also from the azimuthal point of view, we consider the selected shots as the final ones. Otherwise, more shots are added to increase the data coverage.

From the presented four examples in this study, the shot positions have been optimised for three examples (case studies 1-3).
We also use a dataset (case study 4) where the acquisition layout mimics at a smaller scale the classical seismic exploration 3D cross-spread acquisition scheme with orthogonal lines of sources and receivers. This dataset, not being optimised (for a SWT study) will help analysing the criticalities introduced by a non-optimal acquisition scheme.

## 2.2 Estimation of DCs

Once the acquisition layout is finalised, the DCs are estimated from the acquired data. We use a MATLAB code
(Papadopoulou, 2021) that automatically retrieves the DCs between each receiver pair that are collinear with a source.
Here, we provide the general concepts based on which the code estimated the DCs from the raw data (for detail see Papadopoulou, 2021). For each receiver pair, a frequency-domain narrow band-pass Gaussian filter, which was originally proposed by Dziewonski and Hales (1972), is used to analyse the traces into monochromatic components. The traces are then cross-correlated frequency by frequency to produce the cross-correlation matrix. The phase velocities of surface waves
correspond to the maxima on the cross-correlation matrix, but there are many maxima because in the two-station method the observed phase delay is invariant under $2\pi$ translation (Magrini et al., 2022). Hence, to avoid ambiguity in picking the correct maxima, a reference DC is used. The reference DC is estimated automatically using multichannel analysis method (Park et al., 1998) for the positions near to the receiver pair. The code picks all candidate DCs on each cross-correlation matrix. Then, the candidate that is closest to the reference DC at all frequencies is picked. Afterward, a set of QC processes
allow to reject data points that do not follow the smooth trend of the DC and also to remove poor quality DCs.
For the frequency band of the generic $i^{th}$ estimated DC, we put the corresponding phase velocity values into a vector ($\mathbf{V_i}$).
The input data for the inversion is a vector ($\mathbf{d_{obs}}$) containing the phase velocities of all DCs as:

$$\mathbf{d_{obs}} = [\mathbf{V_1}; \cdots; \mathbf{V_i}; \cdots; \mathbf{V_N}] \tag{1}$$

where $N$ is the total number of estimated DCs. The proposed equation by Passeri (2019) is used to approximate the standard
deviation ($\sigma_{V_j}$) of generic $j^{th}$ element of the phase velocity vector ($V_j$) at its corresponding frequency ($f_j$) as:

$$\sigma_{V_j} = \left[ 0.2822\, e^{-0.1819 f_j} + 0.0226\, e^{0.0077 f_j} \right] * V_j, \tag{2}$$

## 2.3 1D forward modelling

We carry out our experiments in a Cartesian coordinate system. The subsurface is discretized into a set of 3D grid blocks where it is assumed that the only unknown parameter of each grid block is the *VS* value while Poisson ratio *(v)* and density
*(ρ)* are assumed to be known as a priori information. Several 1D model points $\mathbf{m}$ are defined and for each one, the DC is computed using a Haskell (1953) and Thomson (1950) forward model modified by Dunkin (1965). For the $k^{th}$ model point $\mathbf{m_k}$, the local phase velocity for a given frequency $V_k(f)$ is determined as:

$$V_k(f) = g(\mathbf{m_k}) \tag{3}$$

**2.4 Computation of forward response**

To obtain the forward response in the curved-ray SWT, first the ray path between the generic receiver pair $R_1$-$R_2$ for each frequency of the DC should be computed. Having computed the phase velocities as a function frequency at the position of each model point, the 2D phase slowness maps at each frequency are built. Then, ray tracing (Noble et al., 2014) is performed at every built phase velocity map to compute the frequency dependent ray paths between the receiver pair ($l_{R_1R_2}$).

To evaluate the accuracy of the ray-tracing algorithm, we have applied it in a to a homogeneous media and noticed that the
error (i.e., deviation from straight-line) in this condition is almost zero (not shown here). The frequency dependent ray path between the receiver pair is discretized to many points. At the generic $i^{\text{th}}$ discretised point along the path, the phase slowness ($p_i$) is computed using a bilinear interpolation among the computed 1D local forward models at the four surrounding model points as:

$$p_i(f) = \frac{\sum_{m=1}^{2} |x_i - x_{m+2}||y_i - y_{m+2}|p_m + |x_i - x_m||y_i - y_m|p_{m+2}}{|(x_1 - x_2)(y_1 - y_2)|} \tag{4}$$

where x and y show the position of each point. Figure 1 shows a schematic representation of the bilinear interpolation along the discretised path between the receiver pair.

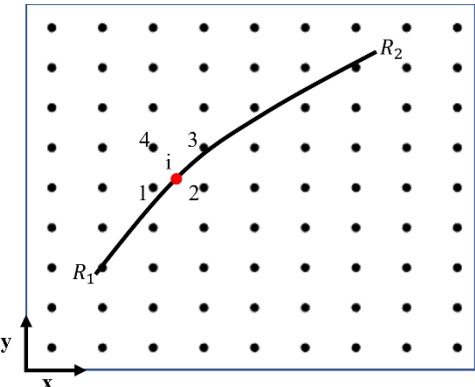

**Figure 1.** The ray path between receivers $R_1$ and $R_2$ at a generic frequency is represented by the solid black line. The phase slowness for any discretised point (i) along the path is computed based on the values of its four adjacent grid points using a bilinear interpolation (Eq.
4). (See also Boiero, 2009).

The path-average phase slowness along the path for each frequency ($p_{R_1R_2}(f)$) is then computed as:

$$p_{R_1R_2}(f) = \frac{\int_{l_{R_1R_2}} p_i(f,l)\,dl}{\int_{l_{R_1R_2}} dl} \tag{5}$$

The corresponding phase velocity ($V_{R_1R_2}(f)$) is equal to the inverse of the computed phase slowness:

$$V_{R_1R_2}(f) = \frac{1}{p_{R_1R_2}(f)} \tag{6}$$

We obtain the vector of the simulated DC for the receiver pair ($R_1$- $R_2$) by:

$$\mathbf{DC}_i = \left[ V_{R_1R_2}(f_1); \ ...; \ V_{R_1R_2}(f_n) \right] \tag{7}$$

where $f_1$, ..., $f_n$ represent the frequency components of the corresponding DC. It should be noted that each estimated DC may have a frequency band different from the others and therefore, the lengths of DCs are not necessarily the same. The experimental DCs consist of a vector of frequencies ($\mathbf{f}$) with their corresponding phase velocities. The vector of the forward

response of the model ($\mathbf{fw}(\mathbf{m})$), which contains the simulated phase velocities corresponding to each element of $\mathbf{f}$, is then obtained as:

$$\mathbf{fw}(\mathbf{m}) = \left[ \mathbf{DC_1}; ...; \mathbf{DC_i}; ...; \mathbf{DC_N} \right] \tag{8}$$

## 2.5 Inversion algorithm

The employed inversion algorithm is based on the method proposed by Boiero (2009). We solve the inverse problem aiming

at minimising the misfit function ($\Phi$) which is defined as:

$$\Phi = \left[ \left( \mathbf{d_{obs}} - \mathbf{fw}(\mathbf{m}) \right)^T \mathbf{C_{obs}^{-1}} \left( \mathbf{d_{obs}} - \mathbf{fw}(\mathbf{m}) \right) \right] + \left[ \left( \mathbf{R_p m} \right)^T \mathbf{C_{R_p}^{-1}} \left( \mathbf{R_p m} \right) \right] \tag{9}$$

where $\mathbf{m}$ shows the vector of the model parameters, $\mathbf{d_{obs}}$ is the observed data, $\mathbf{fw}(\mathbf{m})$ represents the forward response of the model that is computed from Eq. (8). The spatial regularization matrix $\mathbf{R_p}$ contains values of 1 and -1 for the constrained parameters and zeros elsewhere (see Auken and Christiansen, 2004, for details) and the extent of variation of each model

parameter with respect to its neighbouring cells is controlled by the covariance matrix $\mathbf{C_{R_p}}$. To reduce the impact of spatial regularization on the inversion results, in all four examples in this study, a large value ($10^6$) is assigned to $\mathbf{C_{R_p}}$. It means that the VS difference between the neighbouring cells is constrained to 1000 m/s. The matrix $\mathbf{C_{obs}}$ consists of the uncertainties of the observed data and is obtained as:

$$\mathbf{C_{obs}} = diag\left( \mathbf{\sigma_V}^2 \right) \tag{10}$$

where $\mathbf{\sigma_V}$ is computed using Eq. (2). The defined misfit function (Eq. (9)) is minimised iteratively. At the $n^{th}$ iteration, the current model $\mathbf{m_n}$ is updated as (Boiero, 2009):

$$\mathbf{m_{n+1}} = \mathbf{m_n} + \left( \begin{array}{l} \left[ \mathbf{G^T C_{obs}^{-1} G + R_p^T C_{R_p}^{-1} R_p + \lambda I} \right]^{-1} \\ \times \left[ \mathbf{G^T C_{obs}^{-1} \left( d_{obs} - fw \left( m_n \right) \right) + R_p^T C_{R_p}^{-1} \left( -R_p m_n \right)} \right] \end{array} \right), \tag{11}$$

where $\mathbf{G}$ is the sensitivity matrix of the data and $\lambda$ represents the damping factor (see Marquardt, 1963, for details). Two exit criteria are defined to stop the inversion process. The inversion ends when either the ratio of the values of the misfit function ($\Phi$) from the updated model and the current one ($\Phi_{n+1}/\Phi_n$) is less than 1.0001 or the number of iterations exceeds 35.

To estimate the DCs from raw data, we have used the auto-picking code (Papadopoulou, 2021) in which the DCs are sampled uniformly in frequency. This means that each DC is non-uniformly sampled in terms of wavelength which can drive the inversion algorithms to the shallowest part of the subsurface without any significant updates in the deeper portion of the initial velocity model (Khosro Anjom and Socco, 2019). To address this issue, a wavelength-based weighting scheme was applied in the inversion process to compensate for this non-uniformity (see Khosro Anjom et al., 2021, for details). Hence, the $\mathbf{C_{obs}}$ is modified as:

$$\mathbf{C_{obs}} = \begin{bmatrix} \frac{\sigma_{1,1}^2}{w_{1,1}} & 0 & 0 & 0 & \cdots & 0 \\ 0 & \frac{\sigma_{2,1}^2}{w_{2,1}} & 0 & 0 & \cdots & 0 \\ 0 & 0 & \ddots & 0 & \cdots & 0 \\ 0 & 0 & 0 & \frac{\sigma_{i,j}^2}{w_{i,j}} & \cdots & 0 \\ \vdots & \vdots & \vdots & \vdots & \ddots & \vdots \\ 0 & 0 & 0 & 0 & 0 & \frac{\sigma_{N,1}^2}{w_{N,1}} \end{bmatrix}, \tag{12}$$

where $\sigma_{i,j}$ is the standard deviation of the i[th] data point of the j[th] DC, and $w_{i,j}$ is the corresponding weight that is computed as:

$$w_{i,j} = \frac{\Delta \lambda_{i,j}}{\Delta \lambda_{j,max}}, \tag{13}$$

where $\Delta \lambda_{i,j}$ represents the wavelength difference between the data point $i$ of the j[th] DC and the data point with the smallest wavelength distance from $i$, and $\Delta \lambda_{j,max}$ is the maximum computed wavelength difference for the j[th] DC.

**3 Results**

In this section, we apply straight-ray and curved-ray SWT approaches on four datasets and compare the results. For each example, the straight-ray and curved-ray SWT inversions start from the same initial model. Other inversion parameters ($\mathbf{C_{R_p}}$ and $\lambda$) are also the same for the sake of comparison. It should be noted that only VS values are updated during the

inversion and the other parameters ($h$, $v$, and $\rho$) are fixed. In case of the synthetic examples, the true values of $v$ and $\rho$ are used in the inversion. For the field examples, $v$ and $\rho$ are approximated based on the available a priori information. Having erroneous values of $v$ and $\rho$ can induce errors in the inversion results even though the sensitivity of surface waves to VS is more than $v$ (and way more than $\rho$).

## 3.1 Case study 1: the Blocky model

The Blocky model consists of a sequence of layers with vertically increasing velocity values, surrounding two blocks of velocity anomalies which extend 4 m in horizontal and vertical directions (Fig. 2). The receivers are located in a regular grid with 1 m spacing in an area of 20 m × 20 m (Fig. 2a). Sixteen shots (Fig. 2a) were chosen to generate the raw data after optimising the source positions (explained in Section 2.1). The synthetic data were generated using a 3D- finite difference (FD) code (SOFI3D software described in Bohlen, 2002) and no error has been added to the data. The code is an FD modelling program based on the FD approach described by Virieux (1986) and Levander (1988) with some extensions. It can consider viscoelastic wave propagation effects such as attenuation and dispersion, employ higher order FD operators, apply perfectly matched layer (PML) boundary conditions at the edges of the model, and it works in message passing interface (MPI) parallel environment which reduces the running time of the simulations. The used source is a function of Ricker wavelet with dominant frequency of 40 Hz. To avoid numerical dispersion, the minimum element size is defined in a way to have at least 8 grid-points for the shortest wavelength to model the elastic waves propagation with the interpolation order of 4. To respect the wavelength sampling criteria, a mesh with element size of 0.1 m (in horizontal and vertical dimensions) was defined. To ensure the stability of the simulation, the time stepping was set to 1.0e-5 s to satisfy Courant-Friedrichs-Lewy time stability condition. The geophysical parameters of the Blocky model are reported in Table 1.

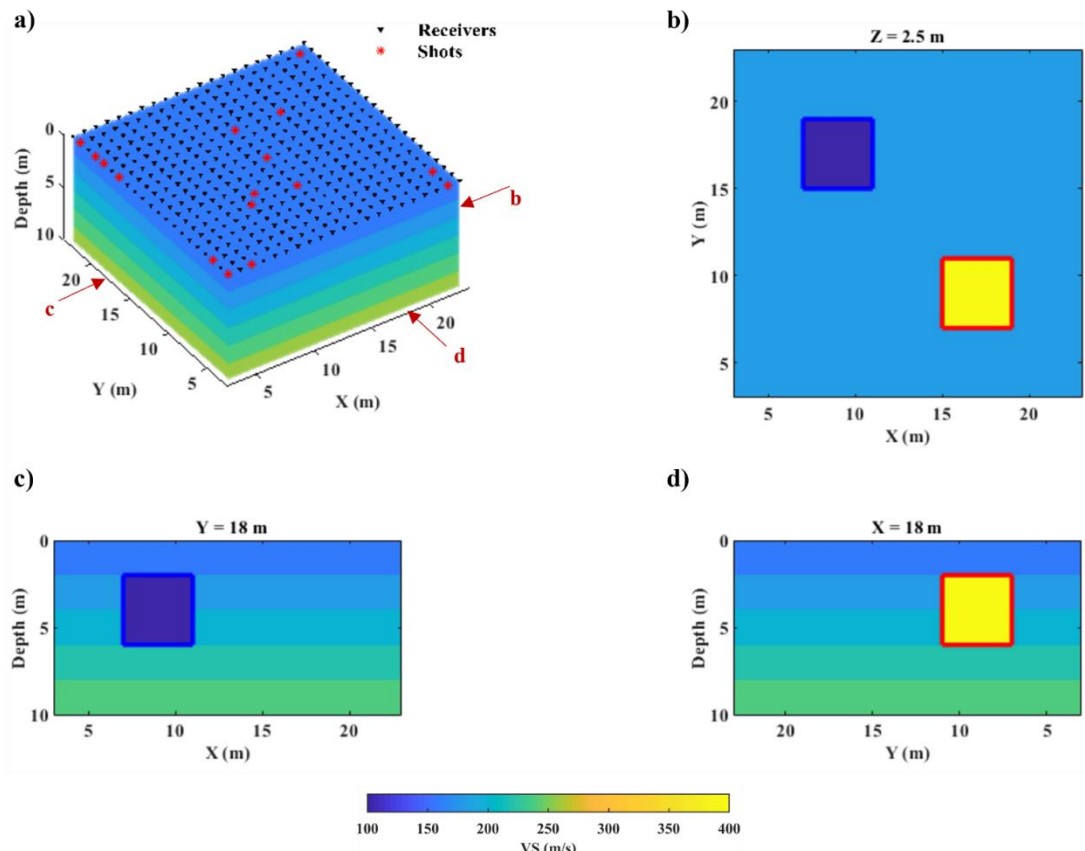

**Figure 2.** True VS model. **(a)** 3D view of the model together with the acquisition geometry. The red arrows and letters represent the location of 2D slices in subplots (b-d). **(b)** Horizontal slice at 2.5 m depth, **(c)** vertical slice at Y=18 m, **(d)** vertical slice at X=18 m. The boundaries of the low- and high-velocity anomalies are superimposed in blue and red, respectively. (X and Y axes do not start from the origin since there is a 3 m absorbing boundary at each side of the model in the simulation)

**Table 1.** Geophysical parameters of the Blocky model.

| Layer | 1 | 2 | 3 | 4 | 5 | Low-velocity block | High-velocity block |
|---|---|---|---|---|---|---|---|
| VS (m/s) | 160 | 180 | 200 | 220 | 240 | 100 | 400 |
| ν (-) | 0.33 | 0.33 | 0.33 | 0.33 | 0.33 | 0.33 | 0.33 |
| h (m) | 2 | 2 | 2 | 2 | 2 | 4 | 4 |
| ρ (kg/m³) | 2000 | 2000 | 2000 | 2000 | 2000 | 2000 | 2000 |

The estimated 971 DCs are shown in Fig. 3a. The initial model for the inversion is defined as a 5-layer 3D model where the thickness of each layer is fixed at 2 m. The horizontal dimensions of each inversion block are 2 m on the side and all blocks have an initial constant VS value of 200 m s$^{-1}$. The Poisson ratio ($v$) and $\rho$ values are assumed to the same as the true model and are equal to 0.33 and 2000 kg m$^{-3}$ in the whole subsurface. The same initial model is used as the starting model for the

SWT inversion in both straight- and curved-ray methods. The inversion ends when the stopping criteria are satisfied. The values of the misfit function at different iterations of the inversions are displayed in Fig. 3b. The VS models at the last 220 iteration of the inversion are displayed in Fig. 4.

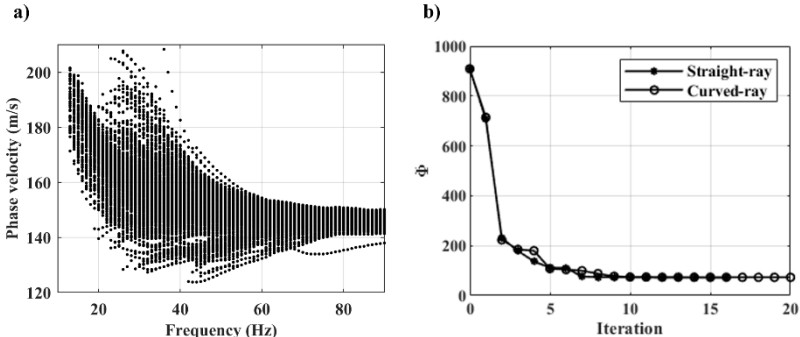

**Figure 3. (a)**The estimated DCs for the Blocky model, **(b)** the values of the misfit function at different iterations.

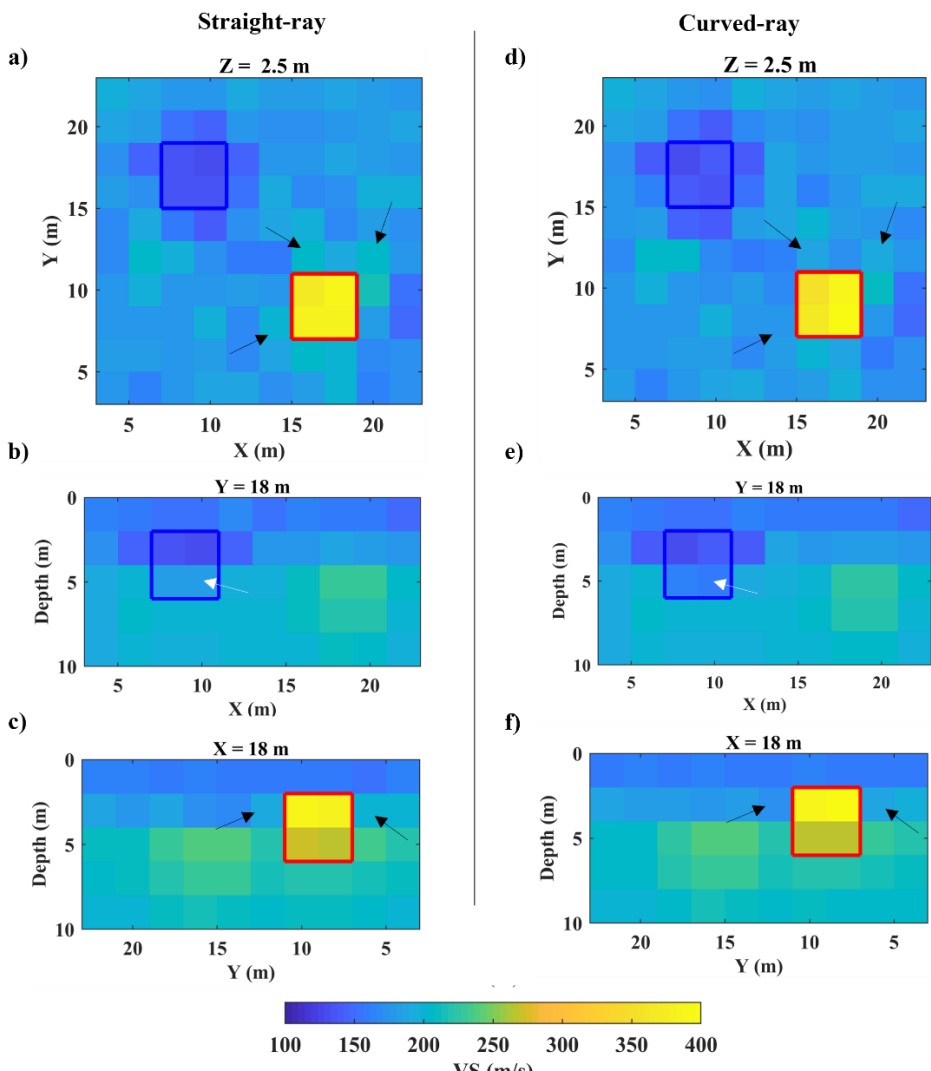

**Figure 4.** The VS models from SWT inversion. Straight-ray SWT results are shown at: **(a)** 2.5 m depth, **(b)** Y=18 m, **(c)** X=18 m, and the
results of the curved-ray for the same slices are displayed at subfigures **(d)** to **(f)**. The boundaries of the low- and high velocity blocks are
superimposed in blue and red. The black and white arrows mark the blocks in which the curved-ray approach provides better results.

Figure 4 shows that straight- and curved-ray SWT have modelled the location and the value of the high-velocity anomaly

quite accurately. The model from the curved-ray method is slightly superior at the grid blocks surrounding the high-velocity

box (the black arrows in Fig. 4). In case of low-velocity anomaly, curved-ray SWT has provided better results, since the

bottom half of the low velocity block is better resolved by the curved-ray approach (the white arrows in Fig. 4b and e). As

shown in Table 4, the curved-ray approach has produced slightly lower model misfit than straight-ray SWT.

## 3.2 Case study 2: the Sand Bar model

The Sand Bar model is designed to simulate a saturated environment where a sand layer is buried in unconsolidated clays. The model contains a curved-shape high-velocity anomaly (the sand layer) embedded between two low-velocity clay layers

235 (Fig. 5). The geophysical parameters of the Sand Bar model are shown in Table 2. The receivers are distributed at the surface as a regular grid with 2 m spacing (Fig. 5a). We defined sources at the same location of receivers and for each source, we computed the aligned receiver pairs. Then, we picked 13 shots (Fig. 5a) which provided the highest data coverage to generate the synthetic data. The same finite difference code used for the Blocky model was used to obtain the Sand Bar synthetic dataset and no error was added to the synthetic data. The source is a function of Ricker wavelet with dominant

240 frequency of 40 Hz, and the minimum element size of the mesh grid was set to 0.1 m to prevent numerical dispersion. The time stepping was equal to 1.0e-5 to avoid simulation instability.

**Table 2.** Geophysical parameters of the Sand Bar model.

| Material | $VS$ (m s$^{-1}$) | $VP$ (m s$^{-1}$) | Computed $v$ | $h$ (m) | $\rho$ (kg m$^{-3}$) |
|---|---|---|---|---|---|
| Clay- 1$^{st}$ layer | 80 | 1700 | 0.497 | 3-6 | 1750 |
| Sand | 150 | 2000 | 0.499 | 3 | 1900 |
| Clay- last layer | 100 | 1850 | 0.498 | 4 | 1950 |

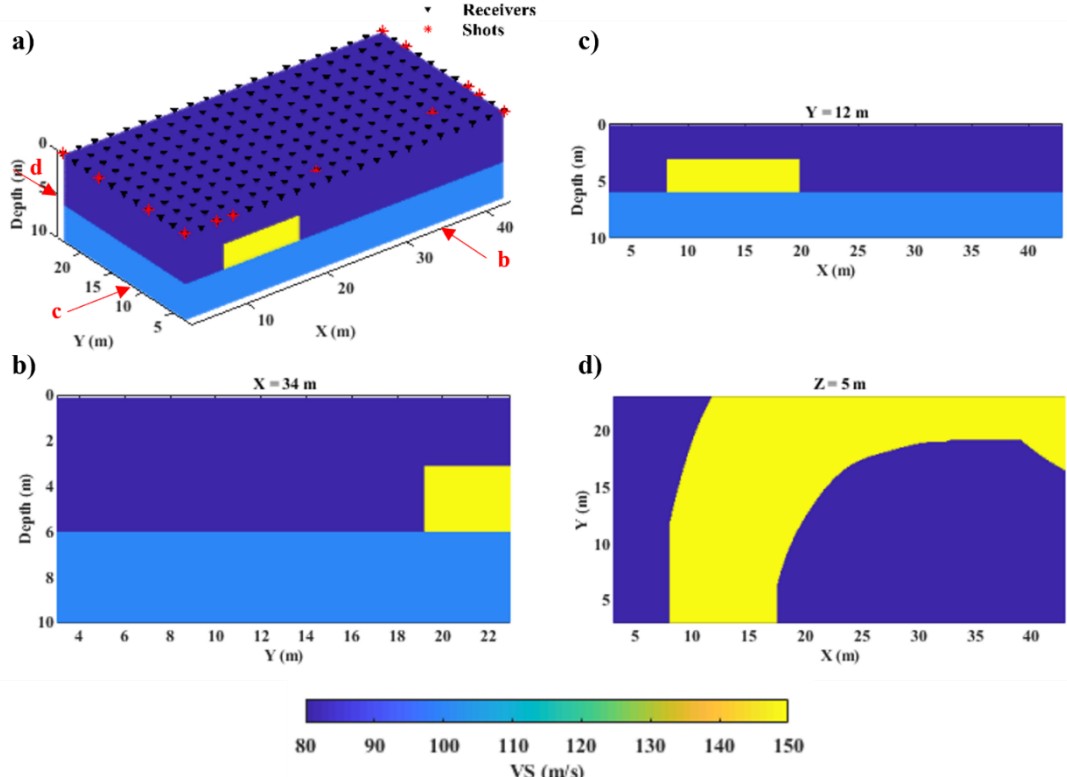

**Figure 5.** True VS model. **(a)** 3D view of the Sand Bar model together with the acquisition geometry. The arrows show the location of the cross-section in the corresponding subfigure. **(b)** Vertical slice at X=34 m, **(c)** vertical slice at Y=12 m, **(d)** horizontal slice at 5 m depth.

The retrieved 1207 DCs are depicted in Fig. 6a. The defined initial model is a 3D model with 10 layers of constant 1 m thickness where the VS values are fixed at 80 m s$^{-1}$. The inversion blocks are 2 m in horizontal dimensions. The initial Poisson ratio and density values are set equal to the values of the true model (Table 2). The misfit function values at different iterations of the inversion are shown in Fig. 6b. The SWT inversion results for one horizontal and two vertical slices of the model are shown in Fig. 7.

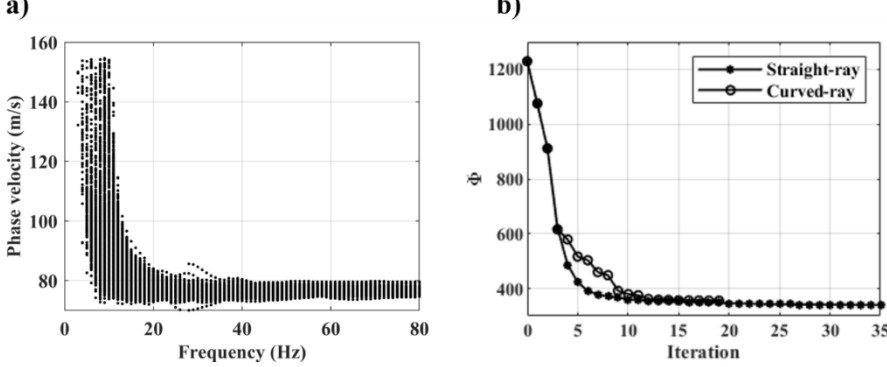

**Figure 6. (a)** The extracted DCs for the Sand Bar model, **(b)** the values of misfit function at different iterations of the inversion.

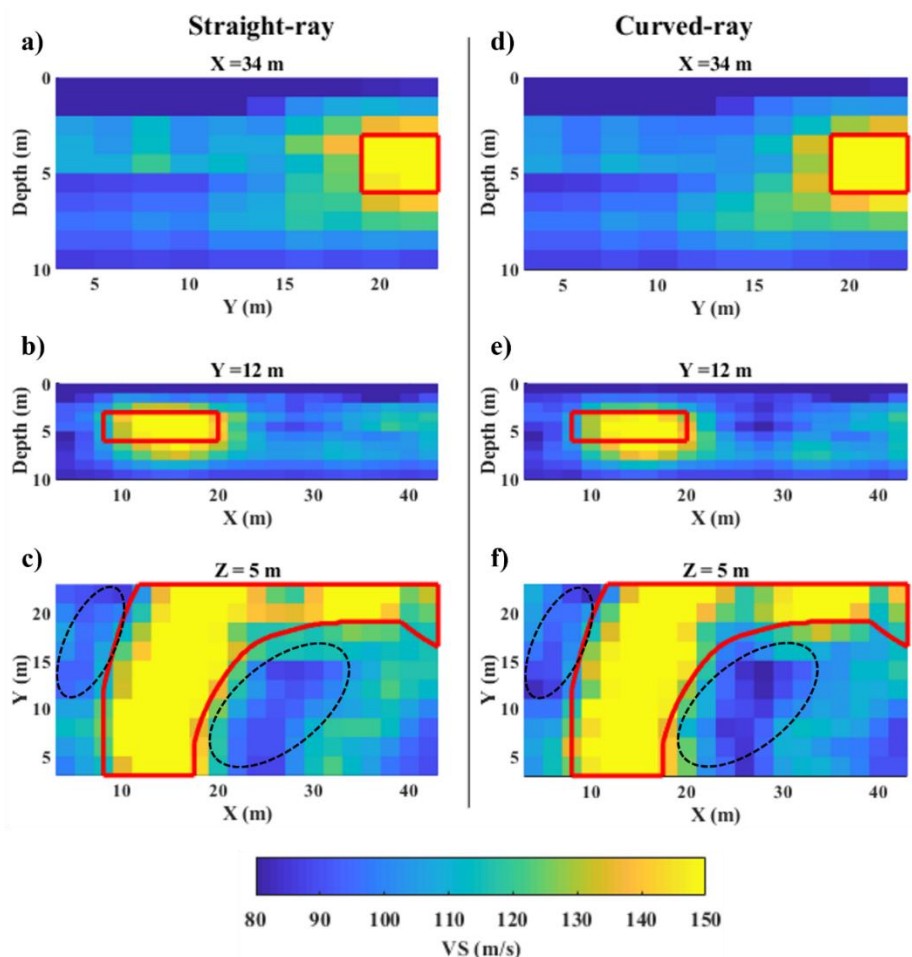

**Figure 7.** SWT inversion results. Straight-ray SWT results are shown at: **(a)** X=34 m, **(b)** Y=12 m, **(c)** Z=5 m, and the results of the curved-ray for the same slices are displayed at subfigures **(d)** to **(f)**. The boundaries of the sand layer are superimposed in red. The corresponding true VS model for each slice is shown in Fig. 4b-d.

We can see in Fig. 7 that the VS models from both approaches are like each other. Figure 7 shows that both methods have successfully located the high-velocity anomaly. Not only the velocity values are close to the true model (Fig. 5b to d), also the shape of the anomaly has been retrieved clearly. The vertical slices at X (Fig. 7a and d) and Y directions (Fig. 7b and e) do not display significant differences. The areas marked in dashed black in the horizontal slices (Fig. 7c and f) shows that the boundaries of the anomaly are slightly clearer in the curved-ray approach (Fig. 7f) and also the VS values in these areas are closer to the true VS value (Fig. 5d).

### 3.3 Case study 3: Pijnacker field

The data were acquired in a field near Pijnacker, South Holland, Netherlands (Fig. 8a). An area of 27 m×30 m was investigated by 120 geophones and 59 shot locations (Fig. 8b). The shot positions were optimised (Section 2.1) for the

locations inside the array area. Fifteen shot locations which provided the highest coverage were chosen as the optimised shot positions. Moreover, 44 shot locations were chosen outside the acquisition area, each one at 3 m distance from every geophone located at the outer boundary of the acquisition area. The source was a vibrator that emitted a linear sweep signal from 2 to 100 Hz for 5 seconds at a force level of 1150 N. Some of the available shallow well data close to the field are depicted in Fig. 8c. They show that the field mainly consists of clay, together with peat and possibly sand in some locations. The extracted972 DCs from the data are displayed in Fig. 8d.

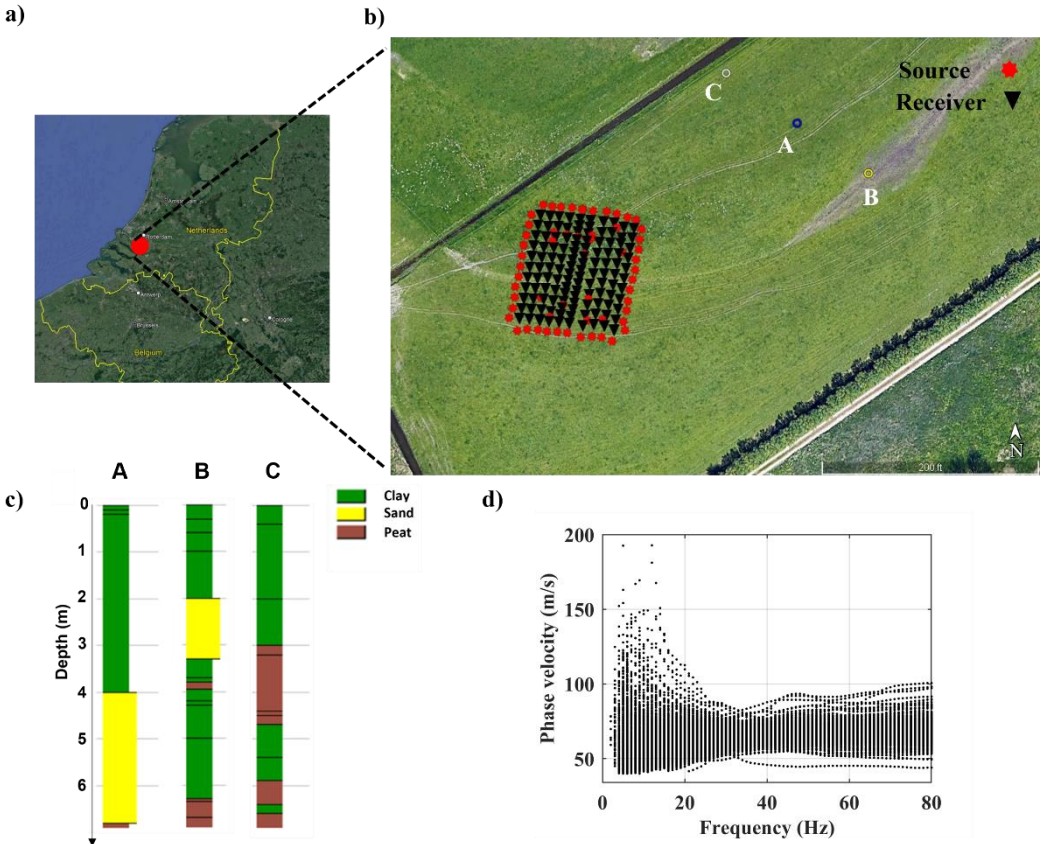

**Figure 8. (a)** Aerial view of Netherlands with the red circle marking the location of the Pijancker field (© Google Earth). **(b)** The acquisition geometry (© Google Earth), **(c)** the available well data near the field. The location of each well is shown with the corresponding letter in subfigure (b). **(d)** The retrieved DCs.

Each inversion block extends 3 m horizontally. In this case, the initial model contains 6 layers where the layers get thicker with depth. The first two layers are 1 m thick, following by two layers of 2 m and two of 3 m. The initial model is defined regardless of the well information. The well data are used later to assess the inversion results. The inversion started from an initial VS value of 60 m s$^{-1}$. Since the medium was (almost) saturated, a high ν value (0.45) was chosen for the initial model. The ρ values in the medium were assumed to be low (1700 kg m$^{-3}$) because it consisted of unconsolidated materials. The

values of the initial model parameters are reported in Table 3. The values of the misfit function at different iterations are displayed in Fig. 9, and the SWT inversion results for straight- and curved-ray methods are depicted in Fig. 10.

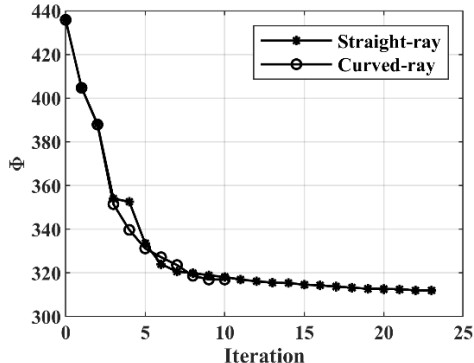

**Figure 9.** The misfit function values at different iterations of SWT inversions.

285

**Table 3.** Parameters of the initial model for the inversion.

| Layer | 1 | 2 | 3 | 4 | 5 | 6 |
|---|---|---|---|---|---|---|
| $VS$ (m s$^{-1}$) | 60 | 60 | 60 | 60 | 60 | 60 |
| $v$ | 0.45 | 0.45 | 0.45 | 0.45 | 0.45 | 0.45 |
| $h$ (m) | 1 | 1 | 2 | 2 | 3 | 3 |
| $\rho$ (kg m$^{-3}$) | 1700 | 1700 | 1700 | 1700 | 1700 | 1700 |

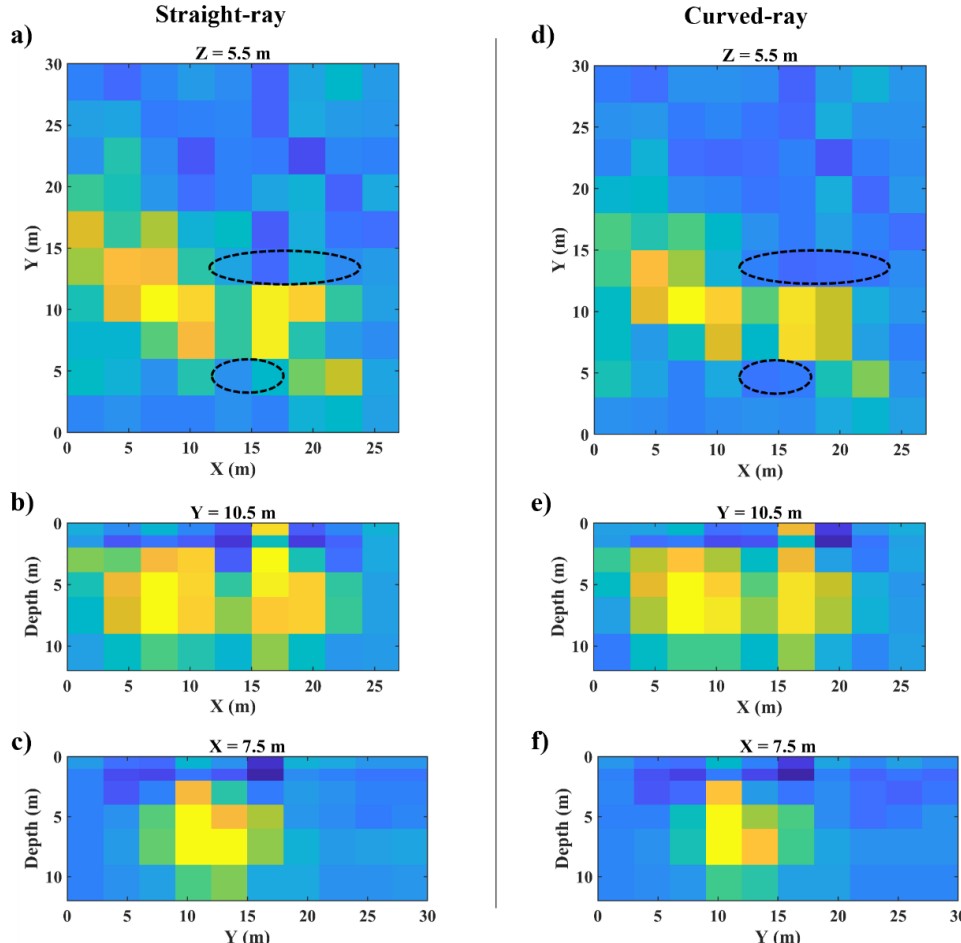

**Figure 10.** SWT inversion results. Straight-ray SWT results are shown at: **(a)** 5.5 m depth, **(b)** Y=10.5 m, **(c)** X=7.5 m, and the results of the curved-ray for the same slices are displayed at subfigures **(d)** to **(f)**. The dashed black marks the blocks in which the VS from the straight- and curved-ray are considerably different.

We can see in Fig. 10 that also in this case the VS models from the straight- and curved-ray SWT are similar. The difference between the horizontal slices (Fig. 10a and d) are clearer than the vertical ones. As shown in dashed black, we can see that the cells around the high velocity portion have lower VS values in the curved-ray (Fig. 10d) than straight-ray (Fig. 10a). A previous 2D full waveform study (Bharadwaj et al., 2015) on a clay-field which was not very far from the field location of our study, obtained a VS model in range of 40-80 m s$^{-1}$ up to 15 m depth. This is in agreement with the inversion results shown in Fig. 10. The high velocity portions relate to the sand. It can be seen in the vertical slices in Fig. 10 that the depth of high-velocity part (sand layer) is mainly in range of 2-9 m which seems reasonable given the a priori well data (Fig. 8c).

### 3.4 Case study 4: CNR field

The field data were acquired at National Research Council (CNR) headquarter in Turin, Italy (Fig. 11a and b). The site consists of compacted sand and gravel formations surrounding an artificial loose sand body. The sand body occupies an area of 5 m × 5 m at the surface and the maximum depth reaches 2.5 m. The receiver layout consists of 4 lines which are 2.5 m apart. Each acquisition line includes 18 vertical 4.5 Hz geophones with 0.5 m spacing. The used source was an 8 kg hammer. The acquisition was carried out in 83 shot locations. Figure 11c shows the acquisition setup which mimics a typical exploration 3D cross-spread acquisition scheme at a smaller scale. The estimated 315 DCs are shown in Fig. 11d.

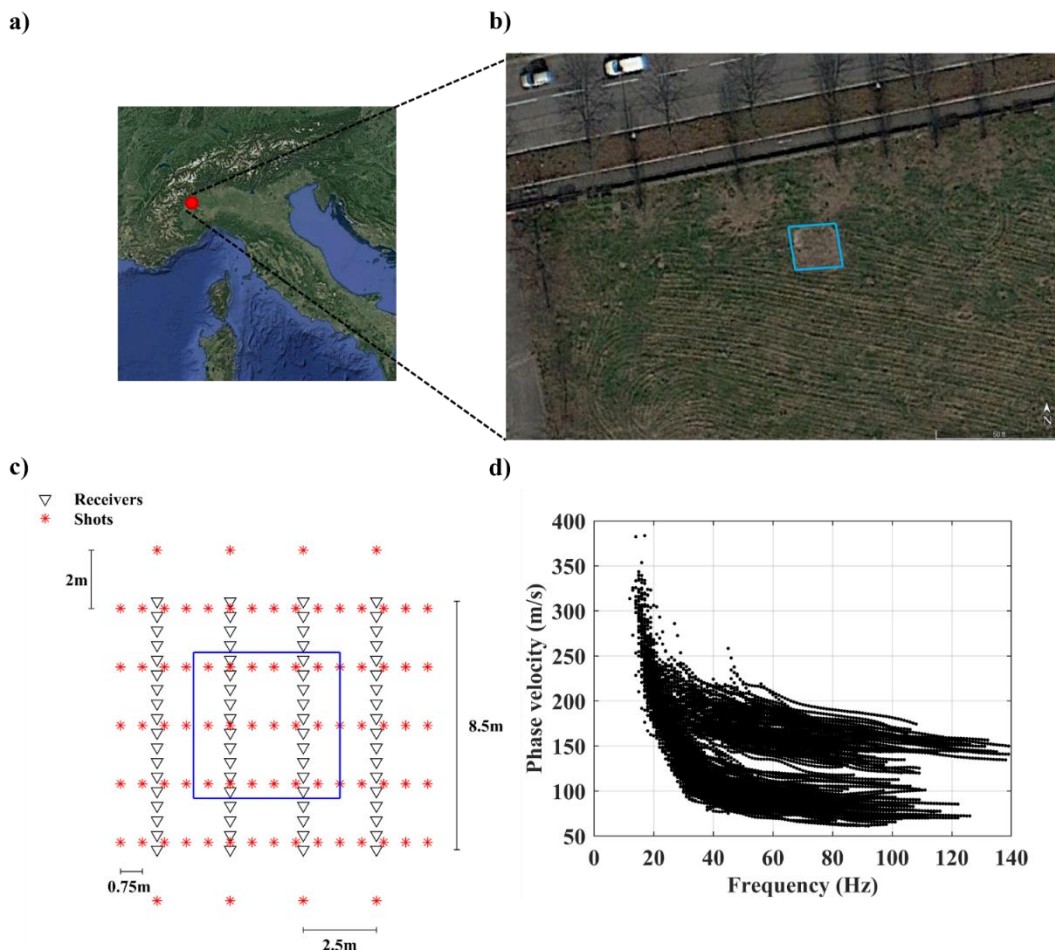

**Figure 11. (a)** Aerial view of Italy with the red circle shows the location of the site (© Google Earth). **(b)** A closer view of the CNR site (© Google Earth). **(c)** The acquisition outline. The boundaries of the sand body are highlighted in blue. **(d)** The estimated DCs.

The inversion started from an 8-layer 3D model where the horizontal and vertical sizes of each grid is 0.5 m, and VS is equal to 200 m s$^{-1}$, $\nu$ is approximated based on a previous study (Khosro Anjom et al., 2019) on the site and fixed at 0.33, and density is fixed at 2000 kg m$^{-3}$ since the site mainly consists of loos sand material. The misfit function values at different

iterations of the inversion are shown in Fig. 12. Both straight- and curved-ray SWT inversion started from the same initial model and the results are presented in Fig. 13.

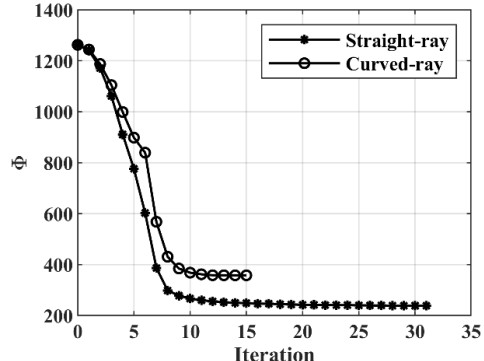

**Figure 12.** The values of misfit function at different iterations of SWT inversions.


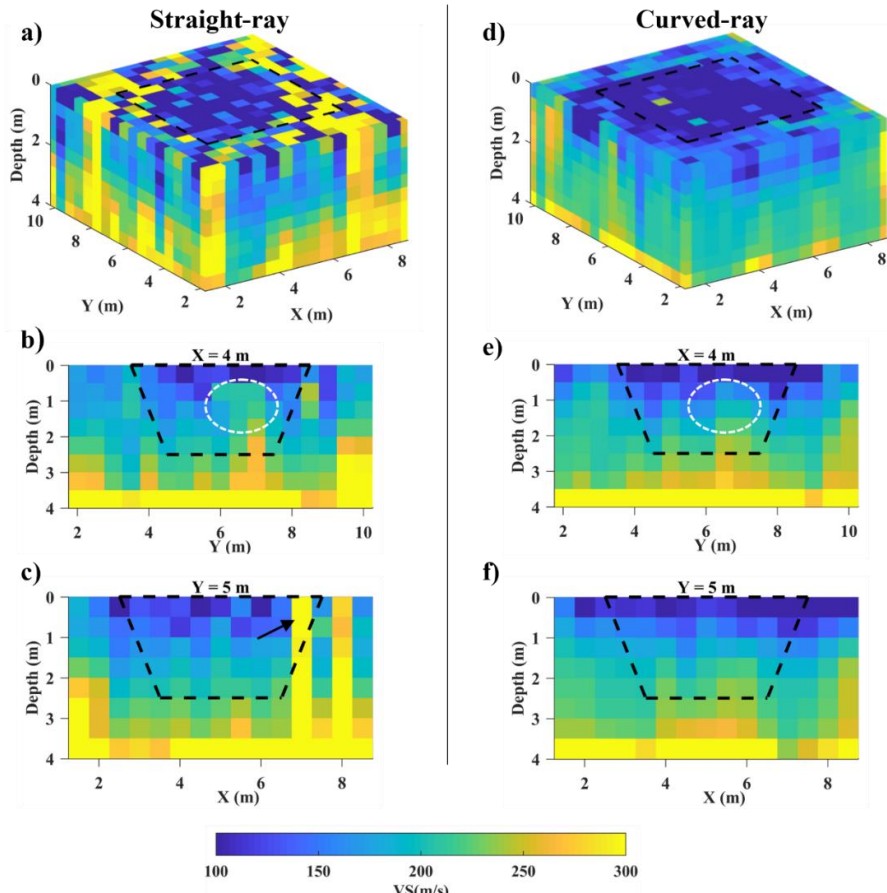

**Figure 13.** The VS models from SWT inversion. The boundaries of the sand body are superimposed in dashed black. The results of the straight-ray are shown as: **(a)** the 3D view of the VS model, **(b)** the vertical slice at X=4 m, **(c)** at Y=5 m, and the corresponding results from the curved-ray SWT inversion are displayed in subfigures **(d)** to **(f)**.

Figure 13 shows that the difference between straight- and curved-ray models are more pronounced in this example. There are some cells with relatively high velocity values inside the sand body in the model obtained from the straight-ray (Fig. 13a). The boundaries of the loose sand body at the surface are better retrieved by the curved-ray SWT (Fig. 10d). The area shown in dashed white in Fig. 13b and e shows that the gradual increase of the VS values with depth inside the sand body is clearer in the model from the curved-ray (Fig. 13e). The black arrow in Fig. 13c shows the high velocity cells inside the loose sand

body in the retrieved model from the straight-ray SWT. The reason is that in this area (close to the interface of the sand body and the background medium) the assumed paths in the straight-ray approach are much shorter than the true paths and therefore, the obtained VS from the inversion becomes unrealistically high. This artefact does not exist in the corresponding slice from the curved-ray SWT (Fig. 13f).

## 4 Discussion

We have applied straight-ray and curved-ray SWT on four datasets and compared the results. In this section, we investigate the results in more details considering ray paths, data weighting, models and data misfits, and computational cost.

### 4.1 Ray paths

The improvement of the model obtained by the curved-ray SWT with respect to straight-ray SWT, particularly at the boundaries of velocity anomalies has been shown in the synthetic and real world examples. Some selected examples of the

computed ray paths at the last iteration of the curved-ray SWT are depicted in Fig. 14.

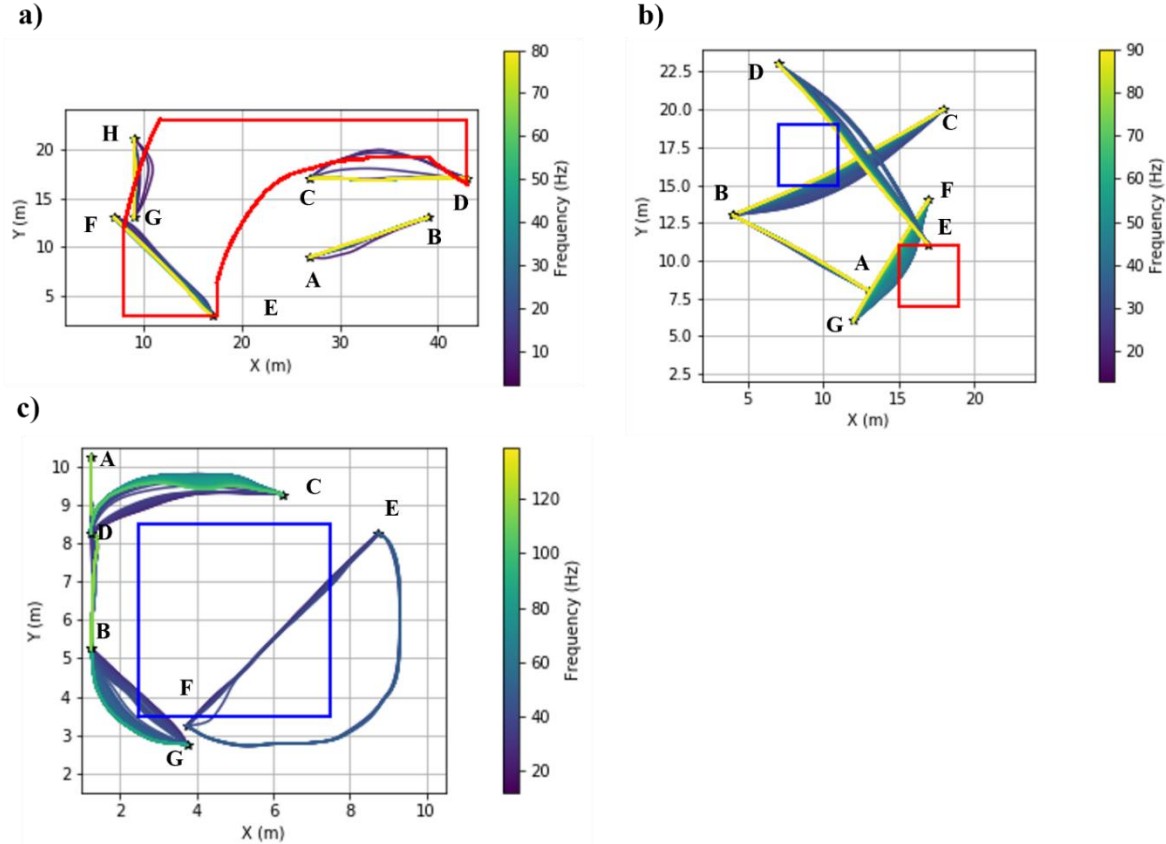

**Figure 14.** Examples of the computed ray paths at the last iteration of the curved-ray SWT inversion for: **(a)** the Sand Bar model, **(b)** the Blocky model, **(c)** the CNR field. The boundaries of the low- and high-velocity anomalies are shown in blue and red, respectively. The receiver locations are labelled as A-H.

In all the three models in Fig. 14 the receivers A and B are located outside the velocity anomalies, and we see that the computed ray paths between them do not cross the anomalies. Therefore, the obtained paths do not deviate considerably from straight lines. In Fig. 14a, the high-velocity anomaly exists at the depth range of 3-6 m. Hence, we can see that the high-frequency components of the DCs, which correspond to the shallow parts of the model, do not deviate from straight

lines. But the lower frequencies (i.e., higher wavelengths) for the C-D and G-H pairs have deviated from straight lines and

travelled through the high-velocity parts. In Fig. 14b, the depth of velocity anomalies is in range of 2 to 6 m. We see that also in this case the ray paths for higher frequencies have almost no deviations from straight lines since they do not cross the anomalies. However, we can see for the obtained paths between B-C and D-E pairs that the lower frequencies have bypassed the low-velocity anomaly. Similarly, lower-frequencies in case of the G-F pair have deviated from straight paths and travelled through the high-velocity anomaly. In Fig. 14c, the sand body (low-velocity anomaly) starts at the surface and

reaches to maximum of 2.5 m depth. Its area shrinks from 5 m × 5 m at the surface to 3 m × 3 m at 2.5 m depth. The shrinkage in size of the anomaly can be seen in the computed path for the B-G, C-D, and E-F pairs, where the degree of the deviation from the straight line decreases as the depth increases (frequency decreases).

Even though the exact boundaries of the anomaly (sand layer) are unknown for the Pijnacker field, the computed ray paths can provide helpful insights. For instance, the computed ray paths from straight- and curved-ray SWT, for the DCs data with

the wavelengths in range of 6-9 m are displayed in Fig. 15.

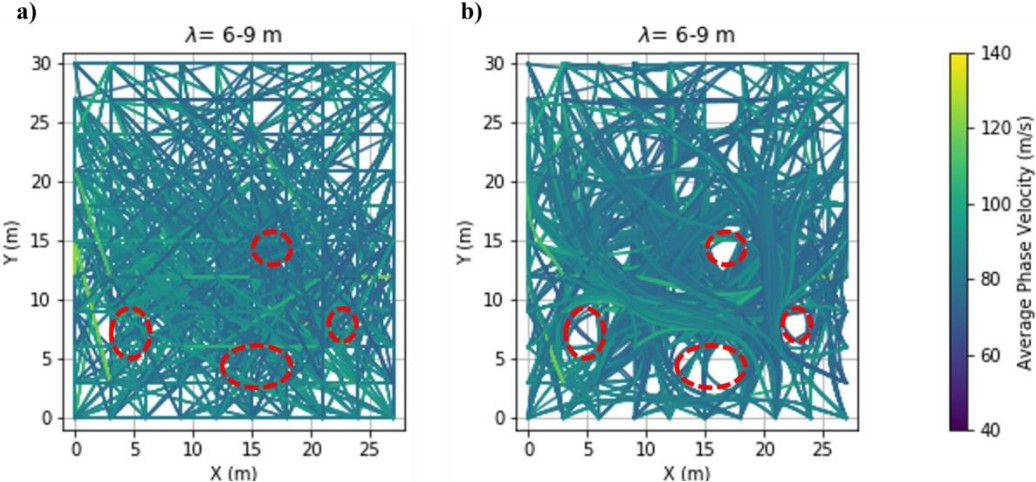

**Figure 15.** The computed ray paths for the data points with wavelengths in range of 6-9 m for the Pijnacker field dataset from: **(a)** the last iteration of the straight-ray, **(b)** the last iteration of curved-ray SWT inversion. The colours of ray paths correspond to the computed path-averaged phase velocity along the path.

Since the initial VS model is vertically and horizontally homogeneous, the initial ray paths for both straight- and curved-ray SWT are straight lines. As shown in Fig. 15a, the ray paths do not change during the inversion in the straight-ray approach. However, the paths are updated at every iteration of the curved-ray SWT inversion. We see that some areas in Fig. 15b (shown in dashed red) are bypassed by almost all the rays even though the data coverage of these areas in the straight-ray approach (Fig. 15a) is considerably high. Therefore, these portions correspond to the low velocity materials, i.e., clay and

peat. The area between these low-velocity portions has both higher concentration of ray paths and higher average phase velocity values. Therefore, they probably show the sand layer. These locations agree with the obtained VS model from the curved-ray SWT inversion (Fig. 10d).

## 4.2 Misfits and computational costs

We have shown the inversion results from the straight-ray and curved-ray SWT. In this part, we compare the results
quantitatively. We carried out the inversions on 40 cores on a cluster with the processor type of Intel® Xeon® E5-2650 v3.
In Table 4, we report the number of iterations ($n_i$), the running time ($r_t$), the maximum memory consumption during the
inversion process ($mem_{max}$), the relative data misfit at the last iteration of the inversion ($e_d$), the relative model misfit for all
cells ($e_m$) and only the ones in the depth range of the target ($e_t$), and the cost of inversion to be run at Microsoft cloud service.
We define the relative data misfit ($e_d$) as:

$$e_d = mean\left(\frac{|\mathbf{d}_{obs} - \mathbf{fw}(\mathbf{m}_{final})|}{\mathbf{d}_{obs}}\right),$$  (14)

where $\mathbf{d}_{obs}$ is the vector of the experimental DCs and $\mathbf{fw}(\mathbf{m}_{final})$ represents the computed forward response of the model at
the final iteration of the inversion. In case of the synthetic examples (the Blocky and Sand Bar models), we can compare the
obtained VS models from the inversion ($VS_{final}$) with the true model ($VS_{true}$). We compute the average relative model misfit
($e_m$) as:

$$e_m = mean\left(\frac{|\mathbf{VS}_{true} - \mathbf{VS}_{final}|}{\mathbf{VS}_{true}}\right),$$  (15)

For each parameter in Table 4, the last column shows the relative difference between the curved-ray and straight-ray
approaches ($d_{CR-SR}$) that has been computed as:

$$d_{CR-SR} = \frac{1}{J}\sum_{j=1}^{J}\left(\frac{a_{CR,j} - a_{SR,j}}{a_{SR,j}}\right)$$  (16)

where $a_{CR,j}$ and $a_{SR,j}$ show the value of each parameter from curved-ray and straight-ray approaches, respectively, and $J$ is the number of examples for which there is an existing value for that parameter.

**Table 4.** The quantitative comparison of straight- and curved-ray SWT.

| SR \ CR | Sand Bar | Blocky | CNR | Pijnacker | $d_{CR-SR}$ |
|---|---|---|---|---|---|
| $n_i$ | 35 / 19 | 16 / 20 | 31 / 15 | 23 / 10 | -18 % |
| $r_t$ (h) | 15.17 / 18.99 | 3.13 / 6.1 | 1.97 / 1.70 | 8.31 / 7.74 | +25 % |
| $mem_{max}$ (GB) | 17.25 / 18.41 | 2.85 / 3.85 | 3.89 / 3.94 | 40.35 / 40.41 | +11 % |
| $e_d$ (%) | 1.119 / 1.121 | 0.985 / 0.987 | 4.40 / 7.21 | 9.25 / 9.81 | +18 % |
| $e_m$ (%) | 15.80 / 15.11 | 8.71 / 8.48 | - | - | -2 % |
| $e_t$ (%) | 23.19 / 20.36 | 9.74 / 9.23 | - | - | -4 % |
| Cost ($) | 64.3 / 80.5 | 13.3 / 25.9 | 8.4 / 7.2 | 35.2 / 32.8 | +25 % |


We can see in Table 4 that in all examples except for the Blocky model, curved-ray SWT has converged in less iterations than the straight-ray. However, the curved-ray SWT has increased $r_t$ by 25 % compared to straight-ray. For all the case studies, curved-ray SWT needed more memory (11 % by average) than straight-ray. In terms of data misfit ($e_d$), straight-ray approach has provided better performance than the curved ray. We can also see that the difference between the obtained $e_d$

values from the straight- and curved-ray SWT is negligible in case of the synthetic examples (the Sand bar and Blocky models), but the difference is more pronounced for the field examples (the CNR and Pijnacker). Despite having higher $e_d$, the curved-ray approach has produced lower model misfits than straight-ray. Using curved-ray SWT has decreased the overall model misfit ($e_m$) and the target model misfit ($e_t$) by 2 % and 4 %, respectively. Finally, we see in Table 4 that using curved-ray SWT has increased the computational cost by an average of 25 %.

**4.3 Impact of the data coverage**

In all the presented examples, the computed VS models from the straight-ray and curved-ray SWT do not differ significantly except for the CNR field. Since the source positions had not been optimised in this example, the DCs coverage is low, particularly in the shallower portion of the medium. Figure 14 depicts the ray paths of the DCs data with the wavelengths in range of 0-1 m.

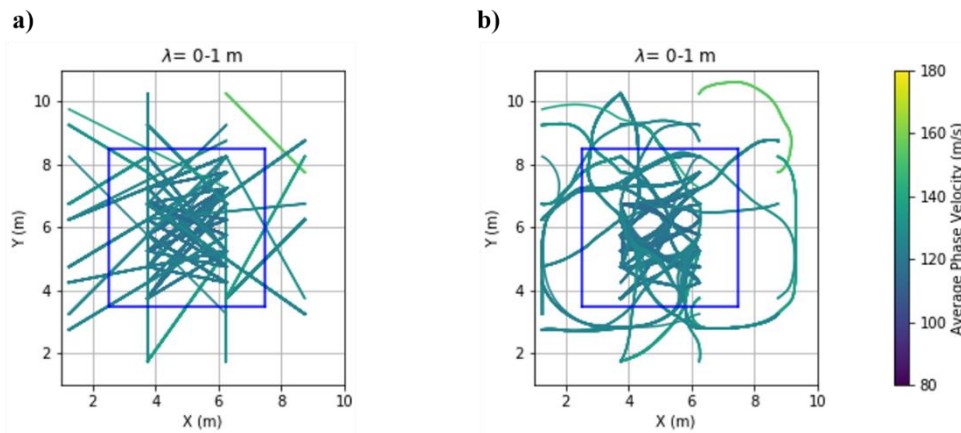

**Figure 16.** The ray paths for the data points of CNR example with wavelengths in range of 0-1 m from: **(a)** the last iteration of the straight-ray, **(b)** the last iteration of curved-ray SWT inversion. The boundaries of the sand body at the surface are superimposed in blue. The colours of ray paths correspond to the computed path-averaged phase velocity along the path.

We can see in Fig. 16a that some areas of the medium are not covered with straight-rays, especially outside the sand body. It should be noted that for both cases, the ray paths at the first iteration are straight lines since the initial model has a constant VS value for all cells. However, in the curved-ray approach (Fig. 16b) the ray paths are flexible and can adjust to the updated subsurface velocity during the inversion process. Figure 16b also shows that the ray paths have been responded properly to the edges of the loose sand body, travelling through the faster part of the model.

### 4.4 Weighting effect

As mentioned previously, uneven sampling of DC data in terms of wavelength can be problematic in SWT inversion. For instance, most of the extracted DC data (81 %) of the Pijnacker field have wavelengths less than 3 m while the available well data from the area suggest that the depth of the target is expected to vary in range of 2-7 m. This can be a serious problem since the inversion might reach the defined stopping criteria without any significant updates in the deeper portion of the initial velocity model. Fig. 17 shows the obtained VS models with (Fig. 17a and b) and without (Fig. 17c and d) the wavelength-based weighting at 5.5 m depth.

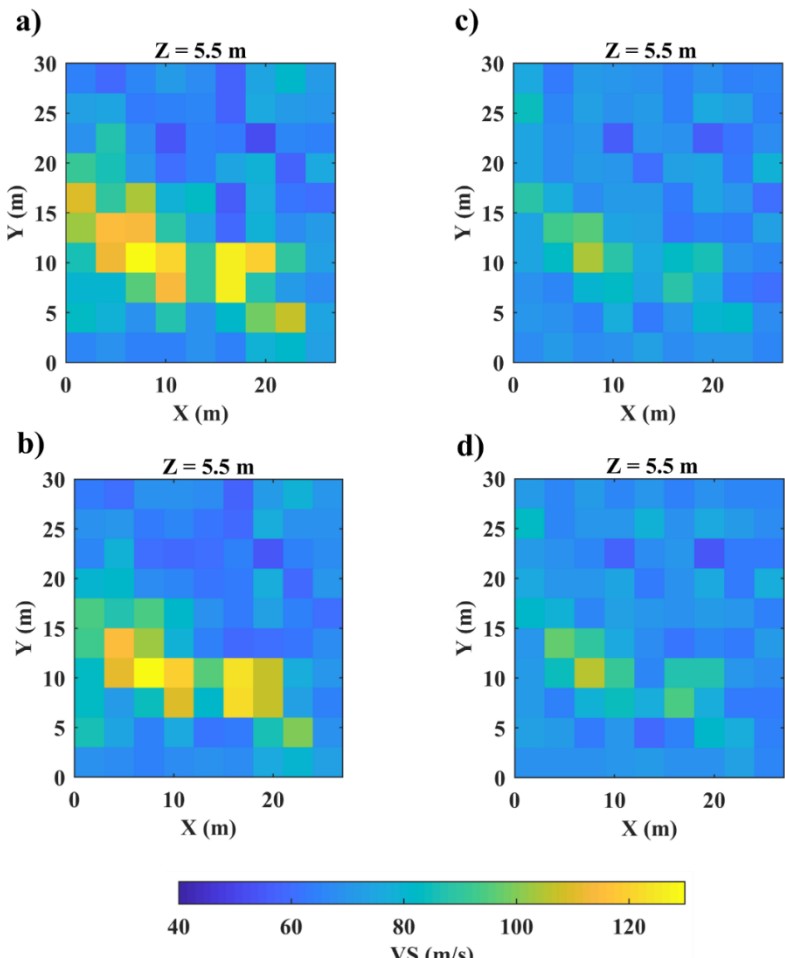

**Figure 17.** The impact of weighting on the SWT inversion results for the Pijnacker dataset. The computed VS model at the depth of 5.5 m from: **(a)** weighted straight-ray, **(b)** weighted curved-ray, **(c)** non-weighted straight-ray, **(d)** non-weighted curved-ray SWT.

In Fig. 17, we can see the improvement of the model after applying the wavelength-based weighting method (Fig. 17a and b) compared with the non-weighted model (Fig. 17c and d) where the non-weighted inversions have barely retrieved any pattern to model the target (sand layer).

## 4.5 Comparison with previous studies

We have shown that optimisation of source positions can provide higher data coverage than a typical 3D cross-spread acquisition scheme. We have evaluated straight-ray and curved-ray at the near-surface scale and comparing our results might not be necessarily agree with previous (global or regional scale) seismological studies. For instance, Spetzler et al. (2001) pointed out that the maximum deviations of ray paths from straight lines is mostly below the estimated resolution. However, we have shown that the deviation from straight lines can be resolved at the near-surface scale (Fig. 14). We also showed that in case of low data coverage, using curved-ray approach significantly improves the obtained VS model. This result might not

agree completely with the result of the (seismological) study by Trampert and Spetzler (2006) where they pointed out that increasing the data coverage is the main factor to increase the resolution of the model. Our results show that in case of high data coverage, the gained model improvement in curved-ray approach may not worth the additional computational effort, which agrees with the result of the study by Bozdag and Trampert (2008).

## 5 Conclusions

We have applied SWT to four datasets and built near-surface VS models. We have compared the obtained results from the straight-ray and curved-ray SWT in terms of data misfit, model misfit, and computational cost. We showed that compared to the straight-ray approach, using curved-ray SWT improves the accuracy of the computed VS model. We illustrated that the acquisition layout can play an important role in the obtained data coverage and consequently in the inversion results. We showed that the classical cross-spread acquisition layout (which was used in the CNR example) may not provide high DC coverage. In this case, the improvement of inversion results from curved-ray SWT can be significant. We also showed that in case of high data coverage, which can be achieved by optimisation of source positions, the difference between the obtained VS models from the straight-ray and curved-ray can be very small even in the presence of high lateral variation of the velocity.

*Code and data availability.* The data may be available by contacting the corresponding author. The code may also be available by contacting the corresponding author after carrying out some additional work to make it user-friendly.

*Author contributions.* MK worked on the processing and inversion of all the datasets, with coordination and supervision of ES and LVS. MK and ES contributed to seismic data acquisition in Pijancker field and LVS coordinated the seismic data acquisition in the CNR site. MK wrote the original paper draft, with contribution from all the authors.

*Competing interests.* The contact author declares that neither him nor his co-authors have any competing interests.

*Acknowledgements*

We would like to thank Seismic Mechatronics for providing the vibrator source, CNR group for giving access to data acquisition, Compagnia di San Paolo for funding the PhD of Mohammadkarim Karimpour, and all the people involved in data acquisition.

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
