# Peer review of "A comparison of straight-ray and curved-ray surface wave tomography approaches at near-surface studies"

_EGUsphere, 2022_

## Author Comment (AC1)

[Figure]

**Figure 1**. *The distribution of the relative model error for the obtained VS model from straight-ray (subfigures a-c) and curved-ray (subfigures d-f) SWT.*

---

## Author Response (AR1)

Dear Editor and Reviewers,

Thank you for your time and revision. We have considered your suggestions to improve the submitted manuscript.

In the following, we provide a point-by-point response for each of your comments.

Best regards,

The Authors

**Responses to the comments of Reviewer#1:**

The submitted manuscript focuses on comparing the straight-ray and curved-ray-based surface wave tomography using four data sets under the near-surface condition. The surface wave tomography technique is a well-established method in regional and global seismology, introducing this technique to near-surface applications could be beneficial for accurately investigating the shallow target, and this topic is within the scope of the journal. However, some points should be addressed before a possible publication. Moreover, there are lots of typos in the manuscript that should be revised properly. The English also needs improvement. I would recommend moderate revision before publication.

*Thank you for your comments and suggestions. We have corrected the typos and provided a response for each of your comments.*

Following, I listed some comments regarding the manuscript.

1. Line 75, Page 3. Here the authors state the Vp and density are assumed to be known as prior information but in the result section (such as in the figure caption of Fig 2) the term "Poisson ratio" is used instead of the Vp. It is recommended to keep the full text consistent.

   *Following this suggestion, we have modified the text and used "Poisson ratio" to keep the full text consistent.*

2. Section 3.1. It is recommended to provide the elastic parameters of the Blocky model to help readers better understand what the model is like.

   *Following this comment, we have provided the elastic parameters of the Blocky model in Table 1 of the revised version.*

3. Line 115, Page 5. It may simply be stated that 16 shots were chosen to generate the raw data after optimized design. The 441 shots may make the readers confusing.

   *We have removed the sentence "To optimize the shot locations, we defined 441 shots …" to avoid confusing the readers. In the revised version, we have added subsection 2.1 to clarify the employed criteria to pick the shot positions.*

4. Line 130, Page 6. What do the red text 'a', 'b', and 'c' and red arrows in Fig 1a represent? They should be demonstrated in the figure caption.

   *We have added the following explanation in the caption to clarify the purpose of using the red arrows "a", "b", and "c":*

   *"The red arrows and letters represent the location of 2D slices in subplots (b-d)."*

5. Line 140, Page 7. "The same initial model is used as the starting model for the SWT inversion in both straight- and curved ray methods". It is recommended to show the readers the iteration curves and inversion parameters (damping factor and weights

used in the regularization matrix). In fact, the inversion parameters of the two inversions should be the same for the sake of comparison.

*We have added the iteration curves for all the examples in Figures 3b, 6b, 9, and 12 of the revised version. We have added the clarifying sentence that all the inversion parameters are kept the same for the sake of comparison in lines 185-187 of the revised version as:*

*"For each example, the straight-ray and curved-ray SWT inversions start from the same initial model. Other inversion parameters ( $\mathbf{C}_{\mathbf{R}_\mathbf{P}}$ and $\lambda$ ) are also the same for the sake of comparison."*

6. Figure 6. For the current version, the red arrow and text are confusing for the readers. It is recommended to look for a better way to show the plot.

*We received similar comment from Reviewer#3 (Fabrizio Magrini). To avoid confusion, we have removed the red arrows from the reconstructed VS models from the inversion (Figures 4, 7, 10, and 13 of the revised version). The intention to put them was to clarify the position of each cross-section but it seemed to add confusion for the readers.*

7. Line 180, Page 12. It is not easy for readers to identify that the boundary in the curved-ray tomogram is clear than the one in the straight-ray tomogram. Providing the model error in this area might be better to support this conclusion.

*Here is the distribution of the model error:*

[Figure]

**Figure 1**. *The distribution of the relative model error for the obtained VS model from straight-ray (subfigures a-c) and curved-ray (subfigures d-f) SWT.*

*As shown is Figure 1, the difference between the model errors from the straight-ray and curved-ray is very small. We have not included this figure in the text of the manuscript because it does not show a clearer difference between straight- and curved-ray approaches than what has been already shown in Fig. 7 of the revised version.*

*We have compared the model errors in Table 4, which shows that curved-ray approach has produced a more accurate model. This agrees with what we had stated in the original manuscript lines 177-178, that is the difference between the VS model from straight-ray and curved-ray approaches in this example is not significant:*

*"The vertical slices at X (Figure 6a and d) and Y directions (Figure 6b and e) do not display significant differences."*

*We have also modified the sentences in line 260-262 of the revised version for clarification:*

*"The areas marked in dashed black in the horizontal slices (**Error! Reference source not found.**c and f) shows that the boundaries of the anomaly are slightly clearer in the curved-ray approach (**Error! Reference source not found.**f) and also the VS values in these areas are closer to the true VS value (**Error! Reference source not found.**d)."*

8. Figure 10. Again, it is recommended to provide the iteration curves and inversion parameters. It seems that the inversion using the straight-ray method becomes unstable and there are some outliers in the tomogram. Do these two methods use the same weights in the smooth regularization?

*We have responded to this comment in the responses to comment#5. We have added the iteration curve for this example. As we explained in the response to comment#5, the inversion parameters are the same for both straight-ray and curved-ray methods. Moreover, the same regularization values (i.e., same weights) have been used for both approaches.*

*As commented correctly by Reviewer#2 (Emanuel Kästle), the observed difference is due to the 'wrong' ray paths in the straight-ray approach. At the edges of the velocity anomaly, the assumed paths by the straight-ray are shorter than the true paths and therefore the velocities are (wrongly) high.*

*We have added the following explanation to the text (lines 324-327 of the revised version) to clarify this issue:*

*"The black arrow in **Error! Reference source not found.**c shows the high velocity cells inside the loose sand body in the retrieved model from the straight-ray SWT. The reason is that in this area (close to the interface of the sand body and the background medium) the assumed paths in the straight-ray approach are much shorter than the true paths and therefore, the obtained VS from the inversion becomes unrealistically high."*

9. For some plots, such as in Figure 12, it is recommended to indicate which data set the plot is related to in the figure caption.

*We have added this clarification in Figures 15-17 of the revised version.*

10. Line 280, Page 20. Please delete the redundant sentence "For each parameter, the values …"

*We have deleted the redundant sentence from the revised version.*

11. Table 3. The abbreviation CR and SR should be mentioned in the context, also, the formula for calculating CR-SR should be demonstrated.

*We have shown the difference between the parameters from curved-ray and straight-ray by $d_{CR-SR}$ and updated Table 4 in the revised version.* We have clarified it in lines 379-383 of the *revised version* as:

"*For each parameter in Table 4, the last column shows the relative difference between the curved-ray and straight-ray approaches ($d_{CR-SR}$) that has been computed as:*

$$d_{CR-SR} = \frac{1}{J} \sum_{j=1}^{J} \left( \frac{a_{CR,j} - a_{SR,j}}{a_{SR,j}} \right)$$

(16)

*where $a_{CR,j}$ and $a_{SR,j}$ show the value of each parameter from curved-ray and straight-ray approaches, respectively, and J is the number of examples for which there is an existing value for that parameter.*"

12. Line 295, Page 21. Again, please delete the redundant sentence "For each parameter, the values …"

*We have removed the redundant sentence.*

13. Please check the sentence above section 4.3: Table 3 that using curved-ray SWT has increases the computational cost by an average of 23 %.

*We have corrected the sentence and deleted the redundant phrase.*

**Responses to the comments of Reviewer#2:**

Dear Editor,

Dear Authors,

the manuscript entitled "A comparison of straight-ray and curved-ray surface wave tomography approaches at near-surface studies" provides an informative study that is certainly of interest to tomographers who consider applying similar methods. The

manuscript is clearly structured and generally well written. The applied methods are appropriate and the authors' conclusions are supported by the results of the two synthetic and of the two real data examples. Nevertheless, there are a couple of points that, in my view, need to be addressed before publication in Solid Earth.

*Thank you for your comments and suggestions. We have replied to all your comments below.*

**Remarks**

In the introduction, a few other studies are mentioned that also apply straight and/or curved-ray tomography, but it is not clear for most of these studies whether they performed a comparison of the thwo approaches. There should be an overview of what the conclusions from other authors on the topic were. I am also missing a (short) discussion at the end of the manuscript on whether the results agree with existing literature.

*We have modified the Introduction and added the following sentences to clarify this issue and provide an overview of what the conclusion from other authors on the topic were in the lines 37-54 of the revised version as:*

*"SWT has been used in seismological studies for decades and different SWT approaches have been compared by seismologists. For instance, Laske (1995) studied deviations from straight line in the propagation of long-period surface waves and concluded that they usually have small effects on the propagation phase. Spetzler et al. (2001) applied both straight-ray and curved-ray SWT methods. They computed the maximum deviations of ray paths from straight lines and pointed out that this maximum is typically below the estimated resolution, except for long paths at short periods. Some studies showed that a more complex forward modelling in SWT did not improve the results (Sieminski et al., 2004; Levshin et al., 2005) while other studies reported obtaining better results (Ritzwoller et al., 2002; Yoshizawa and Kennett, 2004; Zhou et al., 2005). Trampert and Spetzler (2006) pointed out that the choice of regularization has a major impact on SWT results. They studied SWT methods based on ray theory (straight-ray and curved-ray) and scattering theory in which the integral along the ray path is replaced by the integral over an influence zone. They showed that these methods are statistically alike and any model from one method can be obtained by the other one by changing the value of the regularization. They concluded that the only option to increase the resolution of the model is to increase and homogenize the data coverage. Bozdag and Trampert (2008) compared straight-ray and curved-ray SWT methods in their study and mentioned that performing ray tracing could be so time-consuming that the potential gain in crustal corrections on a global scale might not be worth the additional computational effort imposed by ray tracing. Despite seismological studies, a comparison between the performance of straight-ray and curved-ray SWT at the near-surface scale is missing."*

*We have also added Section 4.5 to compare the results of our study with previous studies.*

The abstract needs to be rewritten. It gives a very brief introduction and motivation for the study. But an abstract should summarize the key results of the study. The same applies to the conclusions section which should be rewritten.

One of your key results is that in a scenario with low data coverage, the curved-ray approach performs significantly better. But your synthetic examples do not prove that since you run no example with low coverage. I would suggest that you take one of your two synthetic tests and test whether this conclusion holds.

*We have rewritten the Abstract and Conclusions Sections. We have considered to better highlight the motivation and the key results of the study in the revised version. We have added several lines (14-19 of the revised version) at the end of the Abstract to highlight the key results as:*

*"In three examples we optimise the shot positions to obtain acquisition layout which can produce high coverage of dispersion curves. In the other example, the data have been acquired using a typical seismic exploration 3D acquisition scheme. We show that if the source positions are optimised, the straight-ray can produce S-wave velocity models similar to the curved-ray SWT but with lower computational cost. Otherwise, the improvement of inversion results from curved-ray SWT can be significant."*

*We have rewritten the whole Conclusions Section as well.*

*Regarding the data coverage, we have made several changes to explain this issue more clearly. As we have stated in line 14-18 of the revised version, for three examples in this study (the Blocky model, Sand Bar mode, and Pijnacker example) we have optimized the shot positions to obtain high data coverage. We have devoted subsection 2.1 to describe the employed algorithm to optimize the source positions. For case of the CNR example, the acquisition layout mimics at a smaller scale the classical seismic exploration 3D cross-spread acquisition scheme with orthogonal lines of sources and receivers. This dataset, not being optimised will help analysing the criticalities introduced by a non-optimal acquisition scheme. We believe that the subject of the data coverage has been more clearly explained in the revised version and therefore, we did not repeat a test with one of the synthetic examples with lower data coverage (because the problem is not only about having a low data coverage. It's about the low data coverage that may be obtained from a typical exploration acquisition scheme).*

l. 10    "exact paths" - the term 'exact' is quite vague. In this case you calculate Eikonal paths since they are the solution to the Eikonal equation. These represent an approximation and not necessarily the true/exact paths.

*We have removed the term "exact" and modified the sentence in lines 9-10 to:*

*"Alternatively, curved-ray SWT can be employed by computing the paths between the receiver pairs using a ray-tracing algorithm."*

l. 35    So did Trampert and Spetzler find a difference between the ray-based and the finite-frequency approach?

*We have added the following explanation to show the highlights of the study by Trampert and Spetzler (2006) in lines 44-49 of the revised version:*

*"Trampert and Spetzler (2006) pointed out that the choice of regularization has a major impact on SWT results. They studied SWT methods based on ray theory (straight-ray and curved-ray) and scattering theory in which the integral along the ray path is replaced by the integral over an influence zone. They showed that these methods are statistically alike and any model from one method can be obtained by the other one by changing the value of the regularization. They concluded that the only option to increase the resolution of the model is to increase and homogenize the data coverage."*

l. 38      It sounds like Gouedard did not perform any comparison between straight and Eikonal ray based models. So why cite them here?

*Following this suggestion, that work (Gouedard et al., 2010) has been deleted from the Introduction in the revised version.*

l. 50-56   You present several studies where some applied some form of ray tracing and others didn't. But what is your point? Did these studies find any advantage in ray-tracing? If your point is that researchers have applied different methods to approximate the rays but no-one has done a systematic study, then you should write it like that.

*We have re-written the Introduction and devoted a paragraph to refer to the previous seismological studies where these two approaches were compared. We have clarified in lines 52-53 of the revised version that such comparison is missing at the near-surface scale. This is one of the key points of this study to compare the two methods at the near-surface scale.*

l. 70      "shots are defined as a regular grid" sounds wrong to me. I would rather write something like "shot locations located on a regular grid are tested by calculating the number of aligend receivers for each location."

*Following this comment, we have removed "shots are defined as a regular grid". In the revised version, the Method Section has been expanded largely as suggested by Reviewer#3. We have dedicated subsection 2.1 to explain the employed procedure to pick the shot positions.*

l. 71       How many shots are picked, what are the criteria for the number of picked shot locations?

*As mentioned in the response to the previous comment, we have explained the criteria to pick the shot positions in subsection 2.1.*

l. 71      I assume this approach only applies to receiver layouts on a grid and not in case of irregular/random receiver locations? Maybe you should say so.

*It can be applied also to irregular/ random receiver locations. We have used the guidelines proposed by Da Col et al. (2020). In that study, the receivers are not put as a regular grid.*

*We have clarified it in Section 2.1, lines 87-88 of the revised version:*

*"For a given (random or regular) array configuration, we can optimise the locations of shots to ensure having high coverage DCs with minimum number of shots based on the guidelines by Da Col et al. (2020)."*

l. 75        Which values for Vp and rho are you assuming in your study? How are these values chosen?

*As requested by Reviewer#1, we have changed Vp to Poisson ratio for the sake of consistency. For the synthetic examples, we have used the true values of Poisson ratio and densities, as mentioned in lines 216-217 and 247-248 of the revised version. For the Pijnacker example, we have added the following clarification in lines 279-280 of the revised version as:*

*"Since the medium was (almost) saturated, a high v value (0.45) was chosen for the initial model. The ρ values in the medium were assumed to be low (1700 kg m$^{-3}$) because it consisted of unconsolidated materials."*

*In case of the CNR example, the following clarification has been added in lines 309-310 of the revised version:*

*"… v is approximated based on a previous study (Khosro Anjom et al., 2019) on the site and fixed at 0.33, and density is fixed at 2000 kg m$^{-3}$ since the site mainly consists of loos sand material."*

l. 76-80    I think the way you describe the procedure is a bit complicated. Bascially, you take your 3D model defined by Vs,Vp and rho and extract 1D depth profiles at each point of the model. You then calculate the phase dispersion curve for each profile and join all the dispersion curves to get 2D phase-slowness maps at a set of periods (at how many periods, how do you choose the periods?). The ray tracing is then done in each of the phase-slowness maps separately. I would suggest to rewrite this paragraph.

*We have deleted this paragraph from the manuscript. Instead, we have expanded the Method Section as suggested by Reviewer#3. We have provided more details and clarifications in the Method Section of the revised version.*

l. 99, 102    "uneven sampling", "non-uniform sampling"; it would be helpful to your readers if you could say more precisely what you mean. If I understand correctly, it is that the number of samples is not the same at different wavelengths, i.e. periods?

*We did not mean that the number of samples is not the same at different wavelength. We estimated the DCs in the frequency domain and each DC is sampled uniformly in frequency (delta f is constant). This means that for each DC, the adjacent sampled points have the same $\Delta f$  but not the same  $\Delta \lambda$ . In fact, the portion of the DC with lower wavelength are more sampled than the portion with higher wavelength.*

*We have modified this part of the manuscript and added the following explanation to clarify this subject in lines 172-175 of the revised version:*

*"To estimate the DCs from raw data, we have used the auto-picking code (Papadopoulou, 2021) in which the DCs are sampled uniformly in frequency. This means that each DC is non-uniformly sampled in terms of wavelength which can drive the inversion algorithms to the shallowest part of the subsurface without any significant updates in the deeper portion of the initial velocity model (Khosro Anjom and Socco, 2019)."*

You should also mention how you sample your dispersion curves. From the images, it looks like you have a uniform sampling in frequency. This means that you implicitly put a higher weight on high frequencies (a uniform sampling in period would imply a higher weight on low frequencies). Many researchers therefore apply a log-spaced sampling.

*We have mentioned in lines 172-173 of the revised version that we sample each DC uniformly in frequency:*

*'"… we have used the auto-picking code (Papadopoulou, 2021) in which the DCs are sampled uniformly in frequency."*

*We have added explanations regarding the employed processing tool (the auto-picking code described in Papadopoulou, 2021) and devoted a subsection (2.2) to explain the applied methodology to estimate the DCs from the raw data. As this code samples each DC uniformly in frequency, we impose the wavelength-based weights to increase the weights of the points with lower wavelength.*

l. 105      "sigma_i,j is the standard deviation of the ith data point of the jth dispersion curve". I think there is a mistake in that description. In your matrix, only the trace is non-zero. Your sentence would then imply that for each dispersion curve, you only have a single measurement. Instead, I think that you have several measurements (at a set of periods) for each dispersion curve, so that the measurements from disp curve 1 have standard deviations sigma_1,1 to sigma_n,n, and from disp curve 2 from sigma_n+1,n+1 to sigma_n+m,n+m, and so on...

*In fact, we agree that the Eq. 4 in the original version (equivalent of Eq. (12) in the revised version) could be confusing. To clarify, we have modified the previous equation to:*

$$
\mathbf{C_{obs}} = \begin{bmatrix}
\dfrac{\sigma^2_{1,1}}{w_{1,1}} & 0 & 0 & 0 & \cdots & 0 \\[2ex]
0 & \dfrac{\sigma^2_{2,1}}{w_{2,1}} & 0 & 0 & \cdots & 0 \\[2ex]
0 & 0 & \ddots & 0 & \cdots & 0 \\[2ex]
0 & 0 & 0 & \dfrac{\sigma^2_{i,j}}{w_{i,j}} & \cdots & 0 \\[2ex]
\vdots & \vdots & \vdots & \vdots & \ddots & \vdots \\[2ex]
0 & 0 & 0 & 0 & 0 & \dfrac{\sigma^2_{N,1}}{w_{N,1}}
\end{bmatrix}
$$

l. 109      What is meant by "closest data point"? Please explain. Also, why the delta in delta lambda_j,max if it is the maximum wavelength? (from the delta I would expect a difference)

*The weights in Eq. (13) of the revised version, are computed separately for each DC. For each point of the DC, a wavelength value can be computed based on its phase velocity and frequency. For the generic $i^{th}$ point of the $j^{th}$ DC, the "closest data point" is defined as the point from which the $i^{th}$ point has the smallest wavelength distance. $\Delta\lambda_{j,\max}$ represents the maximum computed wavelength difference for the $j^{th}$ DC. We have clarified these terms in lines 182-183 of the revised version as:*

*"where $\Delta\lambda_{i,j}$ represents the wavelength difference between the data point i of the $j^{th}$ DC and the data point with the smallest wavelength distance from i, and $\Delta\lambda_{j,\max}$ is the maximum computed wavelength difference for the $j^{th}$ DC."*

l. 110       Did you add any error to your synthetic measurements? Please mention in the text.

*We have not added any error to the synthetic data. We have clarified it in lines 197 and 239 of the revised version.*

l. 150       It would be good to give a more quantitative measure for the quality of reconstruction, fo example by providing the variance reduction or simply the misfit to the input model. (I just saw that you did in table 3, I would suggest that you write down these values here or refer to table 3).

*We have added the reference to Table 3.*

l. 199       Are the values of nu and rho in your Table 2 fixed during the inversion? Please mention somewhere in the text. What influence do you expect from the potential errors in these values?

*Yes, the values of nu and rho are fixed during the inversion. We have explained it in lines 187-191 of the revised version:*

*"It should be noted that only VS values are updated during the inversion and the other parameters (h, v, and ρ) are fixed. In case of the synthetic examples, the true values of v and ρ are used in the inversion. For the field examples, v and ρ are approximated based on the available a priori information. Having erroneous values of v and ρ can induce errors in the inversion results even though the sensitivity of surface waves to VS is more than v (and way more than ρ)."*

l. 214       What value for the data standard deviation (sigma, eq 4) do you assume in your synthetic tests and in the real data examples? Is sigma individually determined for each measurement?

*We have added Eq. (2) from which the standard deviation of phase velocities are computed in lines 114-116 of the revised version:*

*The proposed equation by Passeri (2019) is used to approximate the standard deviation (*$\sigma_{V_j}$*) of generic j$^{th}$ element of the phase velocity vector (*$V_j$*) at its corresponding frequency (*$f_j$*) as:*

$$\sigma_{V_j} = \left[ 0.2822 \, e^{-0.1819 f_j} + 0.0226 \, e^{0.0077 f_j} \right] * V_j$$

(2)

l 324      You weighting is based on the wavelength of the signals, but at the same time you argue with the lower number of data at long wavelengths. So should the weight not rather be based on the number of measurements at each frequency? Or, putting the question differently, if I have a dataset with exactly the same number of measurements at each frequency (as is probably the case in your synthetic experiment), do I still need the weighting?

*The weighting is different for each DC. The weighting is not based on the number of measurements at each frequency but rather based on the wavelength of each point of a DC. So, even in case of synthetic example, each DC is non-uniformly sampled in terms of wavelength and the wavelength-based weighting can be applied to compensate this non-uniformity. We believe this concept has been better explained in the lines 172-176 of the revised version as:*

*"To estimate the DCs from raw data, we have used the auto-picking code (Papadopoulou, 2021) in which the DCs are sampled uniformly in frequency. This means that each DC is non-uniformly sampled in terms of wavelength which can drive the inversion algorithms to the shallowest part of the subsurface without any significant updates in the deeper portion of the initial velocity model (Khosro Anjom and Socco, 2019). To address this issue, a wavelength-based weighting scheme was applied in the inversion process to compensate for this non-uniformity (see Khosro Anjom et al., 2021, for details)."*

I would like to refer again to my previous comment on the importance of the sampling of your dispersion curves. If you use a uniform sampling in frequency, it is clear to me that the low frequency measurements are underweighted. Maybe run a test with log sampling and compare to the results in Fig. 14.

*As we have explained in the subsection 2.2 of the revised version, we have used a processing tool uniformly samples the DCs in frequency. We believe that with the provided clarifying explanations on the procedure of weighting in the revised version of the manuscript, the process is described much clearer than the original version. We did not run a test with log sampling because as explained earlier, our processing tool samples DCs uniformly in frequency.*

Fig. 7      There should be a scale on (b). Why is the panel in (d) cropped? It seems that some dispersion curves go also to values slower than 50 m/s.
*We have added the scale for the Pijnacker's acquisition scheme (Fig. 8b of the revised version). We have modified Figure 8b and decreased the lower limit of the y-axis to avoid confusion.*

**Responses to the comments of Reviewer#3**

Dear Editor,

The manuscript "A comparison of straight-ray and curved-ray surface wave tomography approaches at near-surface studies" compares the effectiveness of "straight-ray" and "curved-ray" tomography in seismic imaging tasks. The pros and cons of these two approaches can be considered very well known to the seismological community, but I am not aware of any technical study that discusses them thoroughly and systematically. I suppose that doing so was the purpose of this study.

Unfortunately, I believe that a study of this kind might only be considered useful/appropriate/worth of publication if the technicalities inherent to the methods employed are very clearly explained; this is also very important, of course, to make the work reproducible. The manuscript, however, lacks many details, and does not allow one to fully understand or reproduce the work that was carried out. I suggest that the authors revise their manuscript extensively in this sense. Moreover, the quality of English and of the figures could and should be greatly improved.

The authors will also find my detailed comments/questions below.

Best regards,

Fabrizio Magrini

*Thank you for your suggestions. We have added more details, particularly in the Method Section, to allow the readers to fully understand the applied methodology and increase the reproducibility of the work.*

*We have copied your comments below and assigned a number to each of them. We have provided a response for every comment.*

**Comment#1: Abstract:** This is too general. It does not display any highlights on the results presented in the paper

*Response#1: As also suggested by Reviewer#2, we have modified the Abstract and better highlighted the key results of the study.*

**Comment#2: Introduction:** In general, two different strategies exist to "convert" surface-wave dispersion curves to a 3-D Vs model. As explained in the manuscript, the first (i) involves the calculation of phase-velocity maps, which are then converted to Vs by carrying out many 1-D inversions. The second, instead, (ii) allows one to invert the dispersion curves directly for the 3-D model. In both cases, the data kernels can be calculated (a) by assuming that the waves travel along the great-circle path connecting a given station pair (ray-theory) or (b) by accounting for ray bending (ray-tracing).

In this study, you carry out a comparison between strategies (a) and (b). Is there a reason why you don't contemplate strategies (i) and (ii)? This would hugely benefit both the paper and the seismological community. Moreover, from the introduction, it is not clear which strategy between (i) and (ii) you intend to focus on, and why.

*Response#2: As you mentioned, this study focuses on the comparison between strategies (a) and (b). The comparison between strategies (i) and (ii) is definitely very interesting and can be a topic for a separate study because it is out of the scope of this work. We have reconstructed 3D VS models by direct inversion of DCs. We have clarified it in lines 79-80 of the revised version of the manuscript as:*

*"For each dataset, 3D VS models from straight- and curved-ray SWT are obtained by direct inversion of DCs, …"*

**Comment#3:** I feel that the motivation for the study should be discussed more in-depth, because the pros and cons of ray-theory vs. ray tracing are well known (for reviews, see Rawlinson & Sambridge 2003, Rawlinson et al. 2010).

*Response#3: We have re-written some paragraphs of the Introduction for clarification. The main purpose of this study is to evaluate the performance of straight-ray and curved-ray SWT at the near-surface scale. As stated in lines 52-53 of the revised version of the manuscript, despite seismological studies, such comparison is missing at the near-surface scale. We have also compared our results with previous seismological studies in Section 4.5 of the revised version of the manuscript, and showed that the previous findings in seismology might not be valid at the near-surface scale. Moreover, we have explained the applied criteria to optimize the shot positions for a SWT study using active seismic data at the near-surface scale. This is a key difference between the available seismological studies and near-surface studies using active seismic data. We show the importance of the shot optimization in the obtained dispersion curves coverage and consequently the obtained VS models from the inversions in the Results Section.*

**Comment#4: General Consideration:** This study focuses on very shallow crustal structures; I wonder if the iterative nonlinear-inversion scheme used to convert the dispersion curves to the 3-D Vs model can be considered appropriate in this sense. Wouldn't a globally optimized algorithm be more suited to the solution of a (possibly highly) nonlinear problem such as that of this kind? Adding some consideration on the matter would be useful.

*Response#4: We had stated in lines 59-61 of the original version that:*

*"The computational efficiency is of great importance in seismic near-surface since, compared to seismological studies, the abundance of data at active seismic near-surface projects can increase the computational cost significantly."*

*The computational cost is a key factor for professionals particularly in the near-surface studies with high amount of data. We had shown in Table 3 of the original version, the cost of the SWT using the employed deterministic approach which can be as much of 80 $. This number can increase drastically by using stochastic methods.*

*Moreover, we have shown in Table 4 of the revised version of the manuscript that the final VS model obtained from the iterative non-linear inversion can be quite accurate even though the inversions started from homogeneous initial models.*

**Comment#5: Method:** The method, and the related assumptions, should be explained more clearly (see also the points below). In principle, since this is supposed to be a technical work, I believe this part should be highly detailed, so as to make your work reproducible. I suggest that you expand largely this section, and possibly subdivide it into several subsections. It would be good to explain (i) the calculation of dispersion curves, (ii) the ray-theory vs. the ray-tracing algorithm (with the latter one meriting more consideration, especially if you implemented it on your own), (iii) the forward solver that allows you to measure predicted data from a given Vs model, and (iv) the inversion for Vs. In (iv), it would

be good to say something about the calculation of sensitivity kernels at different periods. Other points are found below.

*Response#5: Following this comment, we have largely expanded the Method Section in the revised version of the manuscript and added way more details and equations. We have also divided this section into:*

*2.1 Optimisation of source layout*

*2.2 Estimation of DCs*

*2.3 1D forward modelling*

*2.4 Computation of forward response*

*2.5 Inversion algorithm*

*We have supported each subsection by related explanations and Equations.*

**Comment#6:** - One can only guess that the inversion strategy chosen by the authors involves the direct inversion of surface-wave velocity for the 3-D structure.

*Response#6: We have clarified it in lines 79-80 of the revised version of the manuscript:*

*"For each dataset, 3D VS models from straight- and curved-ray SWT are obtained by direct inversion of DCs, …"*

**Comment#7:** - Synthetic tests to verify the accuracy of the ray-tracing algorithm should be presented. What is the relative error as a function of distance from the source based on a *homogenous* medium?

*Response#7: To test the accuracy of the ray tracing algorithm, we used a homogeneous medium (the Blocky model with constant VS equal to 200 m/s in the whole medium) and computed the ray paths. As an example, we show the computed paths for a DC where the receivers are located at (19 m, 5 m) and (10 m, 5 m) in the figure below:*

[Figure]

Figure 1. The computed ray paths for different frequency component of the DC with receivers' positions at (19 m, 5 m) and (10 m, 5 m).

*The average relative error of the paths (i.e., deviation from straight-line) is almost zero (7.2e-16). We have not shown this figure in the revised manuscript. Nonetheless, we have added the following clarification in lines 129-130 of the revised version:*

*"To evaluate the accuracy of the ray-tracing algorithm, we have applied it in a to a homogeneous media and noticed that the error (i.e., deviation from straight-line) in this condition is almost zero (not shown here)."*

**Comment#8:** - When you refer to "straight lines", are you referring to great-circle paths? If not, it should be explained that you designed your experiment in a cartesian coordinate system.

*Response#8: We have added the following clarification sentence in line 118 of the revised version of the manuscript:*

*"We carry out our experiments in a Cartesian coordinate system."*

**Comment#9:** - Each vector and matrix in equations (2) and (3) should be thoroughly explained, and their dimensions be explicit. For example, is d_obs your slowness, or is it the arrival time obtained from slowness and ray-path distance? How do you calculate the roughness operator Rp? Generally, the extent of the roughness is determined by a damping scalar coefficient (e.g., Boschi & Dziewonski 1999, Magrini et al. 2022), but you have the matrix C_{R_{p}}. Can you please be more explicit on its calculation? Can you also provide a reference for your equation (3), or alternatively a derivation for it? As it is, it appears different from, e.g., eqs. (51) and (57) of Rawlinson & Sambridge (2003).

*Response#9: We have added the reference for Eq. (3) (equivalent of Eq. 11 in the revised version of the manuscript), which is Boiero (2009).*

*We have extended the Method Section largely and added way more details in this section. We explained in line 112 of the revised version of the manuscript that the dimension of the vector of the experimental data (**d_obs**) is (phase) velocity. We have also clarified in lines 160-162 of the revised version of the manuscript that we have assigned a large value ($10^6$) to the covariance matrix of the spatial regularization matrix:*

*"To reduce the impact of spatial regularization on the inversion results, in all four examples in this study, a large value ($10^6$) is assigned to $C_{Rp}$. It means that the VS difference between the neighbouring cells is constrained to 1000 m/s."*

**Comment#10:** - Is your stopping criterion compatible with previous studies?

*Response#10: In some surface waves studies, the inversion stops when the misfit function reduces less than 1% with respect to the value at the previous iteration (e.g., Garofalo et al., 2015). Since our computational facilities have been improved compared to before, we have defined a lower threshold in our study (0.01%) to make sure that the inversion reaches a local minima. As can be seen in Fig. 6b of the revised version of the manuscript, the misfit value of straight-ray inversion shows a sudden decrease at iteration 27, while the inversion would have stopped at iteration 26 if a higher threshold had been chosen as the stopping criterion.*

**Comment#11: Results:** An important point that does not seem to be discussed is the choice of damping in the two different inversions. Slightly different values of damping can produce slightly different results. I believe it would be important to discuss in some depth this choice, and to demonstrate, to some degree, that the result of your comparisons is not biased by improper use of regularization. (Note that a given value of roughness damping might be ideal for the ray-theory case but not for the ray-tracing case, and vice versa).

*Response#11: We have clarified the choice of regularization 160-162 of the revised version of the manuscript. As pointed out by Trampert and Spetzler (2006), the choice of regularization has a major impact on SWT results. Therefore, we have assigned a very large number ($10^6$) to the regularization values so that the final VS model is not biased by the regularization values and the comparison between straight-ray and curved-ray methods are fair.*

**Comment#12:** - For example, consider Fig 10: I have the feeling that the large differences between the two inversions might derive from the choice of the roughness damping (the straight-ray tomography seems slightly underdamped). Have the authors experimented with different values?

*Response#12: We have provided a detailed response in the response to the comments of Reviewer#1. The observed difference is due to the 'wrong' ray paths in the straight-ray approach. At the edges of the velocity anomaly, the assumed paths by the straight-ray are shorter than the true paths and therefore the velocities are (wrongly) high.*

*All experiments have been done with the same regularization values. As stated in lines 160-162 of the revised version of the manuscript.*

**Comment#13:** - Are your synthetic data generated with the mentioned 3-D finite-difference code only in the Case study 1 or also in the Case study 2? Eventually, a brief explanation of this code could go in the Method section.

*Response#13: The same 3D finite difference code has been used to generate the synthetic data in both case studies 1 and 2. We have added the following sentence in line 238-239 of the revised version of the manuscript for clarification:*

*"The same finite difference code used for the Blocky model was used to obtain the Sand Bar synthetic dataset …"*

*We have added a brief explanation of this code (SOFI3D) in lines 197-201 of the revised version of the manuscript as:*

*"The code is an FD modelling program based on the FD approach described by Virieux (1986) and Levander (1988) with some extensions. It can consider viscoelastic wave propagation effects such as attenuation and dispersion, employ higher order FD operators, 200 apply perfectly matched layer (PML) boundary conditions at the edges of the model, and it works in message passing interface (MPI) parallel environment which reduces the running time of the simulations"*

**Comment#14:** - I am struggling to understand the meaning of the red arrows/letters in Fig. 8, and the caption is not helping me. Probably I am missing something simple, but this suggests to me that the explanation in the caption should be extended or made clearer.

*Response#14: Since Reviewer#1 had the same struggle, we have removed the red arrows from the reconstructed VS models from the inversion (Figures 4, 7, 10, and 13 of the revised version of the manuscript).*

**Comment#15: Discussion:** The relative misfit in equation (6) is a function of **fw(m_final)**. May you please be more explicit on the forward calculation of your dispersion curves (predicted data) based on a given model? Did you use SOFI3D?

*Response#15: No, we have not used SOFI3D for this purpose. Following your previous comments on the Method Section, the process for the forward calculation of the dispersion curves have been thoroughly explained in Section 2.4 of the revised version of the manuscript.*

**Comment#16:-** In the introduction, you refer to Boschi and Dziewonski (1999) while speaking of seismic ambient noise. Clearly, in 1999 ambient-noise tomography did not exist

*Response#16: We have put it its right place, that is at line 24 of the revised version of the manuscript.*

**References**

Nicholas Rawlinson and Malcolm Sambridge. Seismic traveltime tomography of the crust and

lithosphere. Advances in geophysics, 46:81–199, 2003.

Nicholas Rawlinson, S Pozgay, and S Fishwick. Seismic tomography: a window into deep earth.

Physics of the Earth and Planetary Interiors, 178(3-4):101–135, 2010.

Fabrizio Magrini, Sebastian Lauro, Emanuel Kästle, Lapo Boschi. Surface-wave tomography using SeisLib: a Python package for multi-scale seismic imaging, Geophysical Journal International, ggac236, https://doi.org/10.1093/gji/ggac236, 2022

---

## Author Response (AR2)

Dear Editor,

Thank you for your time and revision. We have considered the Reviewers' suggestions to improve the manuscript.

We have attached two versions of the revised manuscript: a copy with the underlined modifications (manuscript_R2_including_corrections) and a cleaned copy (manuscript_R2). We have provided a response for each of the Reviewer's comments in the following.

There are two comments that are connected to each other on which we would kindly request the Editor's decision:

Reviewer#2 in his first comment suggests performing the inversion using one of the presented numerical examples but with lower data coverage to confirm the conclusion that straight-ray tomography works quite well for data acquired with optimised acquisition layout while curved ray provide an improvement in low coverage due to non-optimal acquisition layout. We have hence taken one of the synthetic data and carried out the inversion after decimating the DC to reduce the coverage and included the results in the response to that comment. However, we did not include this result in the manuscript because we think that simply decimating the data to reduce the coverage of a dataset acquired with an optimised acquisition layout would not be equal to having a lower coverage due to a non-optimal acquisition layout, as in the CNR field example. So, we do not feel that this example is very significant and that it adds value to the manuscript. If the Editor decides that it is necessary to include this example also in the manuscript, we would do so.

On the other hand, the last comment of Reviewer#1 is: "Since the highlighted findings in the conclusions section include the influence of acquisition layout and the benefit of curved-ray SWT in un-optimized acquisition layout, providing some additional noise-free tests using a numerical model (similar to the CNR site) might be helpful for convincing the readers. The test could be submitted as supplementary materials".

We think that this request (not mandatory according to the reviewer if we properly understand) would represent a more meaningful example than the one described above, and it would be certainly possible to carry it out. However, we have not provided such new examples since the CNR site is a controlled site with known features and has been used as the benchmark in several previous studies. More importantly, our conclusion from the CNR example is sufficiently soft not to include more studies.

We would like to leave it to Editor to decide whether the presented examples are sufficient, or it is necessary to provide any new numerical example.

Best regards,

The Authors

**Responses to the comments of Reviwer#1:**

It is great to see that the authors revise their manuscript carefully according to the remarks made by reviewers and editors, and the quality of the paper is greatly improved. It could be ready for publication after some minor revisions:

(1) Line 160, Page 6. Since the regularization term in the objective function is the l2-norm of the differences between each neighboring model and the optimization algorithm always tends to minimize the objective function toward zero, the sentence "it means that the vs difference between the neighboring cells is constrained to 1000 m/s" could make the readers confusing.

We have removed the sentence "it means that the vs difference between the neighboring cells is constrained to 1000 m/s" to avoid confusion.

(2) Line 200, Page 8. It might not need to describe the advanced features of the utilized simulation tool (such as viscoelastic) since they are not used in the paper.

Following this suggestion, we have removed the following sentence from the manuscript:

"… consider viscoelastic wave propagation effects such as attenuation and dispersion, …".

(3) Line 395, Page 24. "Figure 14 depicts the ray paths of …", please check the figure number, here it should be figure 16, right?

Correct. We have corrected the reference to the figure.

(4) Since the highlighted findings in the conclusions section include the influence of acquisition layout and the benefit of curved-ray SWT in un-optimized acquisition layout, providing some additional noise-free tests using a numerical model (similar to the CNR site) might be helpful for convincing the readers. The test could be submitted as supplementary materials.

We agree that providing more examples could be helpful for the readers to understand the importance of the source positions optimisation. However, it is not the topic of this paper. At the same time, the CNR site is a controlled site with known features and has been used as the benchmark in several previous studies (Boiero, 2009; Teodor et al., 2017; Khosro Anjom et al., 2019; Hu et al., 2021).

Providing a new numerical example is possible, but it would be a very time-consuming process (takes approximately a few weeks) and cannot be performed within the given time frame for minor revision (one week).

More importantly, as we have mentioned in the Conclusions that "We showed that the classical cross-spread acquisition layout (which was used in the CNR example) may not provide high DC coverage. In this case, the improvement of inversion results from curved-ray SWT can be significant." We think that this conclusion is sufficiently soft to not include more studies. This can be an interesting topic of further investigation.

**References**

Boiero, D.: Surface wave analysis for building shear wave velocity models: Ph.D. thesis, Politecnico di Torino, 2009.

Hu, S., Zhao, Y., Socco, L.V., and Ge, S.: Retrieving 2-D laterally varying structures from multistation surface wave dispersion curves using multiscale window analysis. Geophys. J. Int., **227**(2), 1418-1438, 2021.

Khosro Anjom F., Teodor, D., Comina, C., Brossier, R., Virieux, J., Socco L.V.: Full waveform matching of vp and vs models from surface waves, Geophys. J. Int., **218**, 1873-1891, 2019

Teodor, D., Comina, C., Socco, L.V., Brossier, R., Trinh, P.T., and Virieux, J.: Initial model design for full-waveform inversion—preliminary elastic modeling from surface waves data analysis: in Extended Abstract in 36th GNGTS national convention, 733–756, 2017.

**Responses to the comments of Reviewer#2**

Dear Editor,
Dear Authors,

the manuscript has significantly improved and (almost) all of my comments have been addressed appropriately.
Comment that has not been taken into account
"One of your key results is that in a scenario with low data coverage, the curved-ray approach performs significantly better. But your synthetic examples do not prove that since you run no example with low coverage. I would suggest that you take one of your two synthetic tests and test whether this conclusion holds."
I still think it would be important to properly test this hypothesis. Doing so in a synthetic test would give the reader a better feeling what influence the optimized shot positions have and where to draw the limit between poor and good data coverage. In your conclusions you highlight the importance of optimized shot positions, but is this really the problem in case study 4? If I compare figure 11 and figure 17, it seems like there is no data for many source positions. So even with optimized shot positions, I might run into the same problem of too low data coverage?

Figure 11 (in the previous version of the manuscript) shows the acquisition scheme of the CNR example while Figure 17 (in the previous version of the manuscript) depicts the impact of the data weighting on the inversion results for the Pijnacker example. I assume the reviewer meant "If I compare figure 11 and figure 16" instead of "If I compare figure 11 and figure 17".

The acquisition layout for the CNR example has not been optimized and Figure 16 shows the obtained low data coverage from this set up for the wavelength range of 0-1 m. It is true that there is no data for some shots (by comparing Figures 11 and 16) and the main reason is that no receiver pair is aligned with some shots, and therefore no corresponding DC has been estimated from them. So, it shows the importance of shot position optimization which can prevent this before acquiring the data.

So, to answer the last question (So even with optimized shot positions, I might run into the same problem of too low data coverage?), in the optimization of shot positions the data illumination and

alignment of receiver pairs with each shot are considered and the best shots are picked to generate the data. Therefore, the risk of running into the problem of too low data coverage would significantly decrease.

Moreover, the data coverage is not the only problem here. It is the obtained low data coverage because of using a non-optimized acquisition set up. So, reducing the data coverage for one of the numerical examples might not show the importance of shot positions optimisation and it would not provide a very similar example to the CNR site.

Nonetheless, we have randomly picked 10 % of the original dispersion curves for the Blocky model and carried out the inversions only using those dispersion curves. We have shown the results in the figure below (Fig. 1):

[Figure]

**Figure 1.** The retrieved VS models for the Blocky model with lowered data coverage from: (a-c) straight-ray SWT inversion, (d-f) curved-ray SWT inversion.

In this case, the curved-ray approach has produced a more accurate VS model than the straight-ray method. The obtained relative model error (compared to the true model) from the straight-ray and curved-ray methods are 12.4 % and 9.7 %, respectively.

Additionally, it would be interesting if the results in case study 4 could be made more similar by applying more smoothing in the straight ray example.

Following this suggestion, we have applied stronger smoothing in the straight-ray example. We decreased the values of C_Rp in the straight-ray approach from $10^6$ to 100 and noticed that the VS model has become more similar to the curved-ray approach (with C_R values of $10^6$). We have presented the results in the figure below:

[Figure]

***Figure 2***. *The obtained VS models from: (a-c) straight-ray approach with C_Rp = 100, (d-f) curved-ray approach with C_Rp = $10^6$*

In the matrix in eq. 12, I think the last element on the trace should not have index j=1?

True. We have corrected the last subscripts of the last diagonal element from "N,1" to "N,j".

line 208: I think the parameter "h" appears here for the first time and should be explained.

We have explained the parameter "h" in line 202 of the "manuscript_R2" file.

Table 4: The unit of "Cost" should be time instead of Dollar?

We had already provided the cost in time, i.e., the running time in Table 4. As we had mentioned in the line 373 of the previous version, the cost represents the inversion costs in dollar at the Microsoft cloud service for a certain time and memory. We have provided this parameter to give a better insight for the readers about the actual costs of SWT inversion.

**Responses to the comments of Reviewer#3**

Dear Editor,

Dear Authors,

I found the manuscript of Karimpour et al. much improved, and I am mostly satisfied by the applied corrections. I still believe that the manuscript needs some polishing though, and I suggest that the authors spend one last effort in improving the style of writing and the quality of the English. (And double check for typos.) Besides this, I identified several points, reported below, that I would be happy to see addressed in the revised version of manuscript. Having done so, I will be happy to accept the manuscript for publication.

Sincerely,
Fabrizio Magrini

Notation in mathematical formulas:
I found the use of "DC" (eqs. (7), (8)) and "fw" (eqs. (8), (9), (11)) quite unconventional, i.e. closer to what one would normally write in a computer program rather than in a mathematical formula. I suggest that the authors replace these "symbols" with more conventional ones in their equations. For example, "DC" could be replaced with "c" and "fw(m)" with "g(m)". (Note that the symbol "g" already appears in equation (3), albeit never employed throughout the rest of the manuscript. For this reason, and because equation (3) can be easily and fully explained using plain English, this equation could simply be removed.)

Following this suggestion, we have replaced the symbol "DC" (in eqs. (7), (8) of the previous version) with "c", and "fw(m)" (eqs. (8), (9), (11) of the previous version) with "g(m)". We have also deleted Eq. (3) and explained it in the text.

Dispersion curve vs phase-velocity profile:
The authors employ the term "dispersion curve" (or "DC") to refer to two quite different "objects": (i) the measured -- experimentally or through ray theory -- dispersion curve (which can be seen as the average inter-station phase velocity), and (ii) the 1-D phase-velocity profiles, computed forwardly and associated with each location identified in the xy plane of a 3-D model. To avoid confusion (e.g., paragraph 130), and consistent with many other works (e.g., Magrini et al. 2022), I suggest that the authors use two different terms for the two circumstances, e.g., (i) "inter-station dispersion curves", or simply "dispersion curves", and (ii) "1-D phase-velocity profiles" or "phase-velocity profiles".

To avoid confusion, we have used the term "local DC" for the computed forward model at the position of the model points. We think that using the term "local DC" (instead of "phase velocity profiles") would clarify the concept for the readers and keeps the manuscript consistent with the majority of the surface waves related works in the literature.

We have applied this modification in lines 115-116 of "manuscript_R2" file as:

"For each model point, the **local DC** is computed using a Haskell (1953) and Thomson (1950) forward model modified by Dunkin (1965)."

Paragraph 130:
- Is Vp treated as an unknown? Please clarify

Vp (or equivalently Poisson's ratio) is assumed to be known. We had mentioned it in Section 2.3 of the previous version as:

*"We carry out our experiments in a Cartesian coordinate system. The subsurface is discretized into a set of 3D grid blocks where it is assumed that the only unknown parameter of each grid block is the VS value while Poisson ratio (v) and density (ρ) are assumed to be known as a priori information."*.

- Equation (3) is not very informative: it is simply stating that the kth phase-velocity profile is computed forwardly from the k_th Earth's model parameter. Moreover, the symbol "g" is not employed throughout the rest of the manuscript. I suggest that this equation is removed.

Following this suggestion, we have removed this equation in the new version of the manuscript.

Paragraph 140:
- In response to a comment I made in my first revision, the authors produced a figure of a traced ray (presented in their rebuttal) and wrote in the manuscript "To evaluate the accuracy of the ray-tracing algorithm, we have applied it in a to a homogeneous media and noticed that the error (i.e., deviation from straight-line) in this condition is almost zero (not shown here)." However, I did not find this test satisfactory. The fact that the traced arrivals of a wavefront propagating from the source towards azimuths of ~90°/180°/270°/360° is zero or close to zero is a well known property of ray-tracing algorithms such as fast-marching and fast-sweeping (see, e.g., Fig. 3 in White et al. 2020). What I was suggesting last time is that the authors produce a map of the relative errors between the theoretically predicted arrival times (which are easily computed from a homogeneous medium) and those computed through their ray-tracing algorithm. Something similar to Fig. 3 of White et al. 2020.
- Depending on the result of this test, it might be worth showing the 2-D map of relative errors and spending a few words about it

As suggested, we have provided the map of the relative errors and have shown the results in the figure below:

[Figure]

**Figure 3.** The computed errors of traveltimes for a source at (X, Y) = (13 m, 13 m)

 We have not included this figure in the manuscript because this figure would not be very informative for the readers, and it could be a deviation from the main topic of the manuscript. More importantly, this figure would not add anything new since we have used the work of Noble et al. (2014) in which they have already discussed the traveltimes errors in detail (e.g., Fig.6 of Noble et al., 2014).

Anyhow, we have added the following explanations regarding the possible ray-tracing error in line 121-122 of the "manuscript_R2" file as:

"The accuracy of the employed ray-tracing algorithm has been already discussed by Noble et al. (2014)."

Equation (5):
Is the distance travelled by the wavefront from a source to a receiver calculated through an integral, or, as I suppose, in discrete form through a sum over the lengths of the several segments constituting each path? If you calculate it in discrete form, please rewrite the equation accordingly and/or be explicit about this in the text.

The inter-station paths are discretised. We have rewritten the equation and replaced the integral with the summation. Moreover, we have clarified in the line before the equation that the path is discretised.

"The path-average phase slowness along the **discretised** path for each frequency ($p_{R_1 R_2}(f)$) is …".

Equation (6):
This is probably simply trivial, I suggest that this equation is removed

We have deleted this equation in the newest version of the manuscript.

Paragraph 160: I found this paragraph unclear.
- "The vector of the forward response of the model fw(m)", seems to be a matrix, based on eq. (8)

The forward response is a vector not a matrix. As shown in eq. (8) of the previous version of the manuscript, fw(m) consists of several vectors (DC_i) that are concatenated vertically forming a vector.

- The authors write: "It should be noted that each estimated DC may have a frequency band different from the others and therefore, the lengths of DCs are not necessarily the same", but I am not sure what this implies. Does this mean that the rows of fw(m), each corresponding to a different frequency, have different dimensions depending on the considered inter-station pair? If so, do the authors carry out some sort of interpolation to homogenize the dimensionality of d_obs and fw(m)? Please clarify in the text and eventually rephrase.

Based on eq. (7) of the previous version, the vectors of inter-station phase velocities (DC_i ) are computed and then they are put together in a vector to form the forward response of the model (fw(m) in eq. (8) of the previous version). The dimension of fw(m) and d_obs are essentially the same and no interpolation is needed to be done. We have noticed that the explanations about the different frequency band of each DC might have been confusing to the readers. We have modified the text accordingly to make it more straightforward and understandable as:

*"It should be noted that each estimated DC may have a frequency band different from the others. The vector of the forward response of the model (g(m)), is then obtained as …".*

Reference

Noble, M., Gesret, A., and Belayouni, N.: Accurate 3-D finite difference computation of traveltimes in strongly heterogeneous media, Geophys. J. Int., 199, 422 1572-1585, https://doi.org/10.1093/gji/ggu358, 2014.

REFERENCES
- Magrini et al. 2022. Surface-Wave Tomography of the Central-Western Mediterranean: New Insights Into the Liguro-Provençal and Tyrrhenian Basins. Journal of Geophysical Research: Solid Earth 127.3 (2022): e2021JB023267.
- White et al. 2020. PyKonal: a Python package for solving the eikonal equation in spherical and Cartesian coordinates using the fast marching method. Seismological Research Letters 91.4 (2020): 2378-2389.